# Aggregation and disaggregation features of the human proteome

Tomi A Määttä[1,2], Mandy Rettel[3], Sindhuja Sridharan[1], Dominic Helm[3], Nils Kurzawa[1,2] (iD),
Frank Stein[3] (iD) & Mikhail M Savitski[1,3,*] (iD)

## Abstract

**Protein aggregates have negative implications in disease. While reductionist experiments have increased our understanding of aggregation processes, the systemic view in biological context is still limited. To extend this understanding, we used mass spectrometry-based proteomics to characterize aggregation and disaggregation in human cells after non-lethal heat shock. Aggregation-prone proteins were enriched in nuclear proteins, high proportion of intrinsically disordered regions, high molecular mass, high isoelectric point, and hydrophilic amino acids. During recovery, most aggregating proteins disaggregated with a rate proportional to the aggregation propensity: larger loss in solubility was counteracted by faster disaggregation. High amount of intrinsically disordered regions were associated with faster disaggregation. However, other characteristics enriched in aggregating proteins did not correlate with the disaggregation rates. In addition, we analyzed changes in protein thermal stability after heat shock. Soluble remnants of aggregated proteins were more thermally stable compared with control condition. Therefore, our results provide a rich resource of heat stress-related protein solubility data and can foster further studies related to protein aggregation diseases.**

**Keywords** aggregation; disaggregation; heat shock; proteomics; thermal proteome profiling
**Subject Category** Proteomics
**Mol Syst Biol. (2020) 16: e9500**

## Introduction

Insoluble protein deposits are a hallmark for many devastating neurodegenerative diseases, such as Alzheimer's, Parkinson's, and Huntington's disease (Valastyan & Lindquist, 2014). Understanding the basic principles of protein (mis)folding, (dis)aggregation, and other features of protein quality control is essential when attempting to tackle and interfere with the causes of those diseases.

Mass spectrometry-based proteomics has become an effective tool for unbiased analysis of the effects of cellular perturbations on a system-wide scale (Cox & Mann, 2007). Modern mass spectrometry analysis allows the quantification of thousands of proteins from multiple samples simultaneously by using one of many labeling techniques such as stable isotope labeling by amino acids in cell culture (SILAC), isobaric tags for relative and absolute quantification (iTRAQ), or tandem mass tags (TMT) (Bantscheff et al, 2012). Recently, combination of protein and peptide level labeling, termed hyperplexing (Dephoure & Gygi, 2012), has allowed to quantify proteomes from even tens of samples in one mass spectrometry experiment (Dephoure & Gygi, 2012; Savitski et al, 2018; Aggarwal et al, 2019).

Proteome-wide mass spectrometry-based studies have been previously used to characterize aggregation-prone proteins in different organisms and conditions. These include, for example, aging nematode (David et al, 2010; Reis-Rodrigues et al, 2012; Walther et al, 2015), mice expressing disease-causing mutant of huntingtin protein (Hosp et al, 2017), mice cells exposed to different stress conditions (Sui et al, 2020), and yeast under chemical or heat stress (Ibstedt et al, 2014; O'Connell et al, 2014; Wallace et al, 2015; Weids et al, 2016). Although these studies involved quite different organisms and conditions, some similarities could be found. For example, chaperones and other proteostasis components were enriched in aggregates from aging nematode (David et al, 2010) and mice expressing huntingtin with disease-causing mutation (Hosp et al, 2017). Similarly, chaperones were found in aggregates when yeast cells were exposed to hydrogen peroxide, arsenite, or azetidine-2-carboxylic acid (Weids et al, 2016). Another similarity is the aggregation of ribosomal proteins in stress conditions, such as aging in nematode (Reis-Rodrigues et al, 2012) and heat (Wallace et al, 2015) or arsenite stress (Ibstedt et al, 2014) in yeast. However, ribosomal proteins were also found in aggregates at physiological conditions in yeast (Ibstedt et al, 2014).

Cells have multiple ways to handle protein aggregates (Tyedmers et al, 2010; Miller et al, 2015; Mogk et al, 2018). Irreversibly

---

1 Genome Biology Unit, European Molecular Biology Laboratory, Heidelberg, Germany
2 Faculty of Biosciences, Collaboration for Joint PhD Degree between EMBL and Heidelberg University, Heidelberg, Germany
3 Proteomics Core Facility, European Molecular Biology Laboratory, Heidelberg, Germany
  *Corresponding author. Tel: +49 6221 387 8560; E-mail: mikhail.savitski@embl.de

damaged proteins can be degraded by the ubiquitin-proteasome system (Balchin *et al*, 2016). Larger aggregates can be cleared by autophagy or secreted out from the cells (Tyedmers *et al*, 2010). To maintain protein homeostasis, protein degradation can be balanced by upregulated protein synthesis. In addition, protein disaggregation and re-folding allow to rescue functional proteins from the aggregates (Doyle *et al*, 2013; Mogk *et al*, 2018; Nillegoda *et al*, 2018).

Disaggregation of aggregated proteins was initially observed and characterized in yeast (Parsell *et al*, 1994). A proteome-wide study showed that the disaggregation of heat-induced aggregates is the main strategy for yeast to deal with the aggregates (Wallace *et al*, 2015). However, disaggregation in yeast is conducted by a Hsp100 disaggregase (Sanchez & Lindquist, 1990; Parsell *et al*, 1994) that has no homologue in the human genome (Mosser *et al*, 2004; Shorter, 2011; Doyle *et al*, 2013; Nillegoda & Bukau, 2015). In metazoans (including humans), the disaggregase activity of Hsp100 is likely replaced by a Hsp70 chaperone system (Rampelt *et al*, 2012; Doyle *et al*, 2013; Nillegoda & Bukau, 2015; Nillegoda *et al*, 2015; Mogk *et al*, 2018) which opens the question of how human cells handle aggregates of endogenous proteins.

Here, we studied heat-induced aggregation and disaggregation of endogenous human proteins *in situ*. We developed a hyperplexed quantitative mass spectrometry assay to measure protein solubility after transient non-lethal heat shock and during recovery. The aggregating proteins were enriched in nuclear proteins, intrinsically disordered regions, high molecular mass, high isoelectric point, and hydrophilic amino acids. After characterizing the features of aggregation-prone proteins, we analyzed the dynamics of disaggregation patterns. We found that the majority of aggregating proteins were rescued from the aggregates. The disaggregation rates correlated with the initial loss of solubility in heat shock and with the proportion of disordered regions in the proteins. In addition to aggregating proteins, we analyzed proteins that remained soluble after heat shock by monitoring their changes in thermal stability. Strikingly, non-lethal heat shock triggered thermal stabilization of aggregation-prone proteins invoking immediate protection against aggregation. We also detected changes in thermal stability for a large number of proteins, including proteins related to stress signaling, DNA binding, and protein quality control complexes.

# Results

## Monitoring protein solubility after heat shock and during recovery

Dynamic SILAC labeling (Ong *et al*, 2002; Doherty *et al*, 2009) was used to distinguish between pre-existing proteins and newly synthesized proteins (Fig 1A). K562 human leukemia cells were cultured in light SILAC medium. The medium was switched to heavy SILAC 90 min before heat treatment. We assume that during this period prior to heat treatment, the intracellular pool of arginine and lysine from light SILAC medium would be consumed. That would allow a more accurate quantification of pre-existing proteins since the signal from newly synthesized proteins can be filtered out.

Prior to heat treatment, cells were partitioned into two aliquots which were exposed to either 44°C (heat shock) or 37°C (mock shock) for 10 min (Fig 1A). A control sample was collected before partitioning the cells (pre-shock). The heat shock temperature was chosen so that it did not compromise cell viability (Fig EV1).

After heat treatment, the cells were allowed to recover at 37°C (Fig 1A). Samples were collected directly after heat treatment and during recovery of 1, 2, 3, and 5 h. Samples were lysed with mild detergent (NP-40) to preserve protein aggregates (Reinhard *et al*, 2015), and soluble protein fractions were collected. After tryptic digestion, peptides were labeled with TMT tags (McAlister *et al*, 2012; Werner *et al*, 2012; Fig 1B). Tagged samples were pooled, fractionated offline, and analyzed on an Orbitrap mass spectrometer (Fig 1B) to distinguish newly synthesized (heavy) and pre-existing proteins (light) in the MS1 scan and different experimental conditions in the MS2 scan (Fig 1C).

## Characterization of aggregation-prone proteins

We collected mass spectrometry data for 7,226 proteins. To obtain a high-quality dataset, we required that a protein had to be quantified in all conditions with at least two unique peptides in at least two biological replicates. The resulting high-quality data included 4,786 light-labeled (pre-existing) and 1,269 heavy-labeled (newly synthesized) proteins with high reproducibility (Appendix Figs S1 and S2).

The protein aggregation was analyzed from the pre-existing (light) fraction. Directly after heat shock, the abundance of 300 proteins (< 7% of quantifiable proteome) decreased significantly (Benjamini–Hochberg-adjusted $P$-value < 0.05 in LIMMA analysis and fold change < 2/3) in the soluble fraction when compared with mock-shocked sample (Fig 2A). We refer to those proteins as aggregators (Fig 2A). At the same time, the majority of proteins (4,486) remained soluble after heat shock (Fig 2A, soluble). While aggregators lost intensity in the soluble fraction, the total protein amount remained constant, as estimated from samples lysed with strong detergent (SDS) (Fig EV2A–C). This indicates that the observed loss of solubility is not an artifact of heat-induced degradation.

Based on the readout of this experiment, we cannot state whether a decrease in solubility is caused by formation of amorphous aggregates (Wang *et al*, 2010), structured fibers (Wang *et al*, 2010; Knowles *et al*, 2014; Bauerlein *et al*, 2017), phase separation (Brangwynne *et al*, 2009; Riback *et al*, 2017), or any other homo- or heteromeric (Senohrabkova *et al*, 2019) protein assemblies [with or without other co-assembling biomolecules, such as RNA (Saad *et al*, 2017)]. However, we assume that a decrease in solubility results in formation of an insoluble protein deposit that we from now on simply refer to as aggregation.

Comparison of the ratio of NP-40-extracted (soluble sub-population) and SDS-extracted (total) proteins reports on the solubility status of a protein. Under physiological conditions, proteins involved in phase separated membrane-less nuclear organelles—such as the nucleolus—have been shown to contain an insoluble sub-population (Becher *et al*, 2018; Sridharan *et al*, 2019). Analysis of NP-40/SDS ratio of pre-existing protein pool between pre-shock and heat shock conditions showed that the aggregators included proteins that were completely soluble as well as proteins that have an insoluble sub-population under physiological conditions (Fig EV2D). The extent of loss of solubility after heat shock was comparable between the two types of aggregators (Fig EV2E). We also observed that proteins from the cytosolic ribosome had an insoluble fraction in unstressed conditions (Fig EV2F). Similar

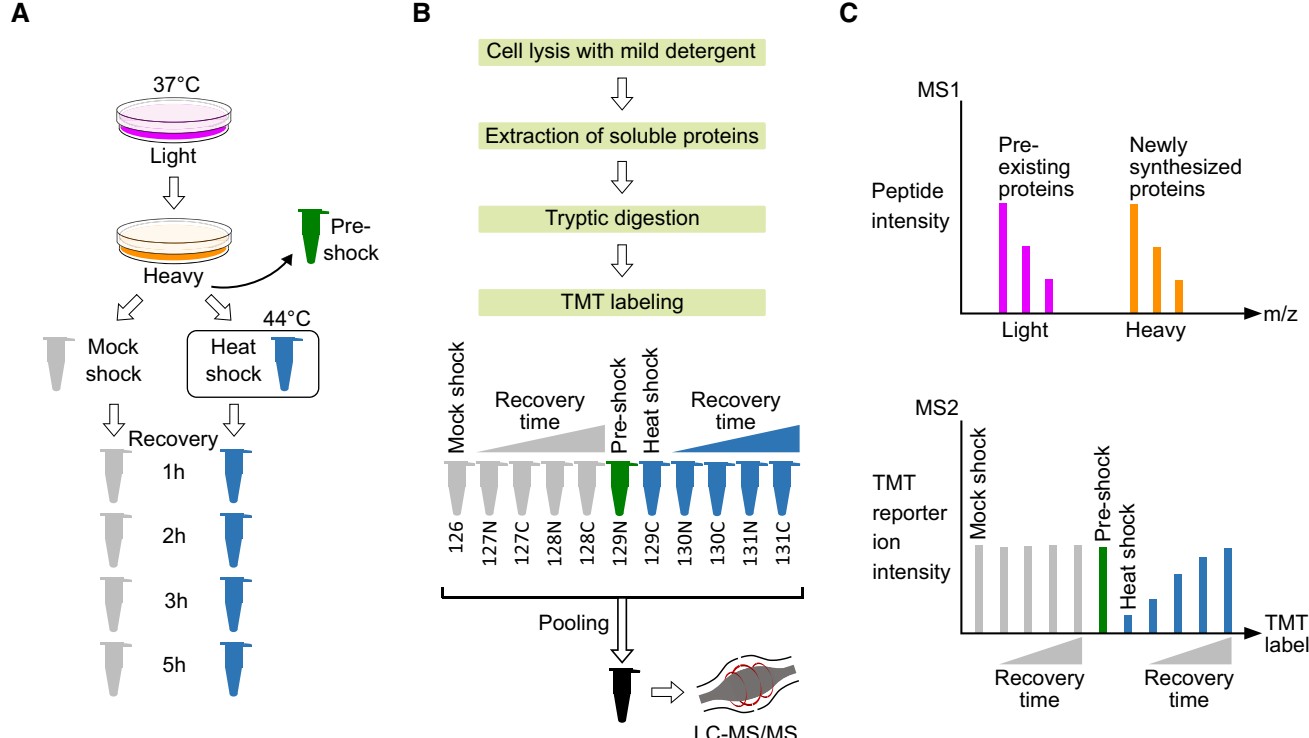

**Figure 1. Experimental design for quantitative proteome-wide solubility measurements after heat shock.**

A  Dynamic SILAC, heat treatment and recovery. K562 human cells were grown at 37°C in light SILAC medium. 90 min before heat treatment, the medium was changed to heavy SILAC containing stable heavy carbon and nitrogen isotopes in arginine and lysine amino acids. These heavy amino acids are incorporated into newly synthesized proteins while the light versions remain in pre-existing proteins. A control sample was collected (pre-shock) prior to partitioning cells for heat treatment. The cells were treated either with heat shock (44°C) or mock shock (37°C) for 10 min. After heat treatment, cells were allowed to recover at 37°C and samples were collected at different time points.

B  Sample processing. Samples were lysed with mild detergent (0.8% NP-40), soluble proteins were extracted and digested to tryptic peptides. Peptides were labeled with TMT labels and pooled. After offline reversed phase fractionation, samples were analyzed with mass spectrometer.

C  Quantification of soluble protein fraction. MS1 scan allows to separate peptides from newly synthesized proteins (heavy) and pre-existing proteins (light). MS2 scan allows for peptide (and later protein) identification and, based on TMT reporter ion intensities, quantification from different samples. For MS2 scan, a hypothetical example is shown for aggregating and disaggregating protein from pre-existing (light) fraction.

observations have been made in yeast (Weids *et al*, 2016). We speculate that this insoluble fraction represents ribosomal proteins in the nucleolus, where the ribosomes are assembled. Contrary to cytosolic ribosomes, we found that proteins from mitochondrial ribosomes are fully soluble in unstressed conditions (Fig EV2F).

To gain a deeper view into the properties of aggregators, we analyzed their physicochemical characteristics (Fig 2B–D). Aggregators were more hydrophilic (Fig 2B; lower gravy score; $P = 1.11 \times 10^{-34}$) and positively charged (Fig 2C; higher isoelectric point; $P = 9.58 \times 10^{-4}$) when compared to proteins that stayed soluble after heat shock. We found a negative correlation (Pearson's $r = -0.73$, $P = 0.00023$) between amino acid hydrophobicity (gravy score) and amino acid composition in aggregators (Fig 2D). The increased isoelectric point was due to enrichment of positively charged arginine and lysine residues in aggregators; the negatively charged residues were either enriched (glutamate) or had similar occurrence in aggregators as in the soluble proteins (aspartate) (Fig 2D).

Next, we looked at structural features of aggregators (Fig 2E–G). Aggregators were enriched in high proportion of intrinsically disordered regions (Fig 2E; $P = 4.51 \times 10^{-40}$) and high molecular weight (Fig 2F; $P = 1.21 \times 10^{-29}$). To expand the structural view, we calculated the fraction of predicted secondary structure elements in proteins (Fig 2G). Aggregators and soluble proteins contained similar amounts of alpha helices ($P = 0.37$) while aggregators contained less beta sheets ($P = 1.88 \times 10^{-12}$). In accordance with the results for disordered regions, aggregators were enriched in (random) coil-like structures (Fig 2G; $P = 1.36 \times 10^{-4}$).

Almost half (> 42%) of aggregators were annotated to be part of a protein complex while the same holds true for only a quarter (< 25%) of soluble protein (Fig 2H, $P = 7.64 \times 10^{-11}$). Protein complexes that had at least 60% of the members aggregating are shown in Fig 2I. All of them (Paf complex, TFIIIC complex, DNA repair complex, and SWI/SNF chromatin-remodeling complex) are nuclear complexes that operate on chromatin. To analyze whether the complexes had truly distinct aggregating parts, we compared the heat shock-induced solubility changes between all members in each complex (heatmaps in Fig 2I). The complexes had similar solubility changes between aggregators that were distinct from the soluble proteins. This suggests that protein complexes composed mainly of

aggregators contain distinct and unstable sub-structures rather than proteins with a continuum of different stabilities. However, the coherent aggregation was not evident when all complexes with at least two aggregators were analyzed (Fig EV2G).

We performed GO term enrichment analysis for the aggregators using all quantified proteins as background. The aggregators were enriched in nuclear proteins involved in DNA binding, chromatin organization, and transcription regulator activity (Fig EV3A). The enrichment of nuclear proteins in aggregators was complementary observed by analyzing protein localization annotations (Fig EV3B); the analysis also indicated that soluble proteins were enriched in cytoplasmic proteins. Since the aggregators are defined from a

comparison between heat- and mock-shocked samples, all detergent-based or other technical biases related to the lysis conditions would be cancelled out.

One feature observed in heat and other stresses is the formation of cytoplasmic stress granules (Collier & Schlesinger, 1986; Collier *et al*, 1988; Ivanov *et al*, 2018). Stress granule-forming factors related to translation have been observed to aggregate in yeast upon heat stress (Grousl *et al*, 2013; Wallace *et al*, 2015). Interestingly, from 44 proteins assigned to a GO term "cytoplasmic stress granule" that we could quantify in our dataset, only two were found to aggregate (TARDBP and RBM4), while the solubility of the other 42 was not affected by the heat shock (Fig EV3C). The different results

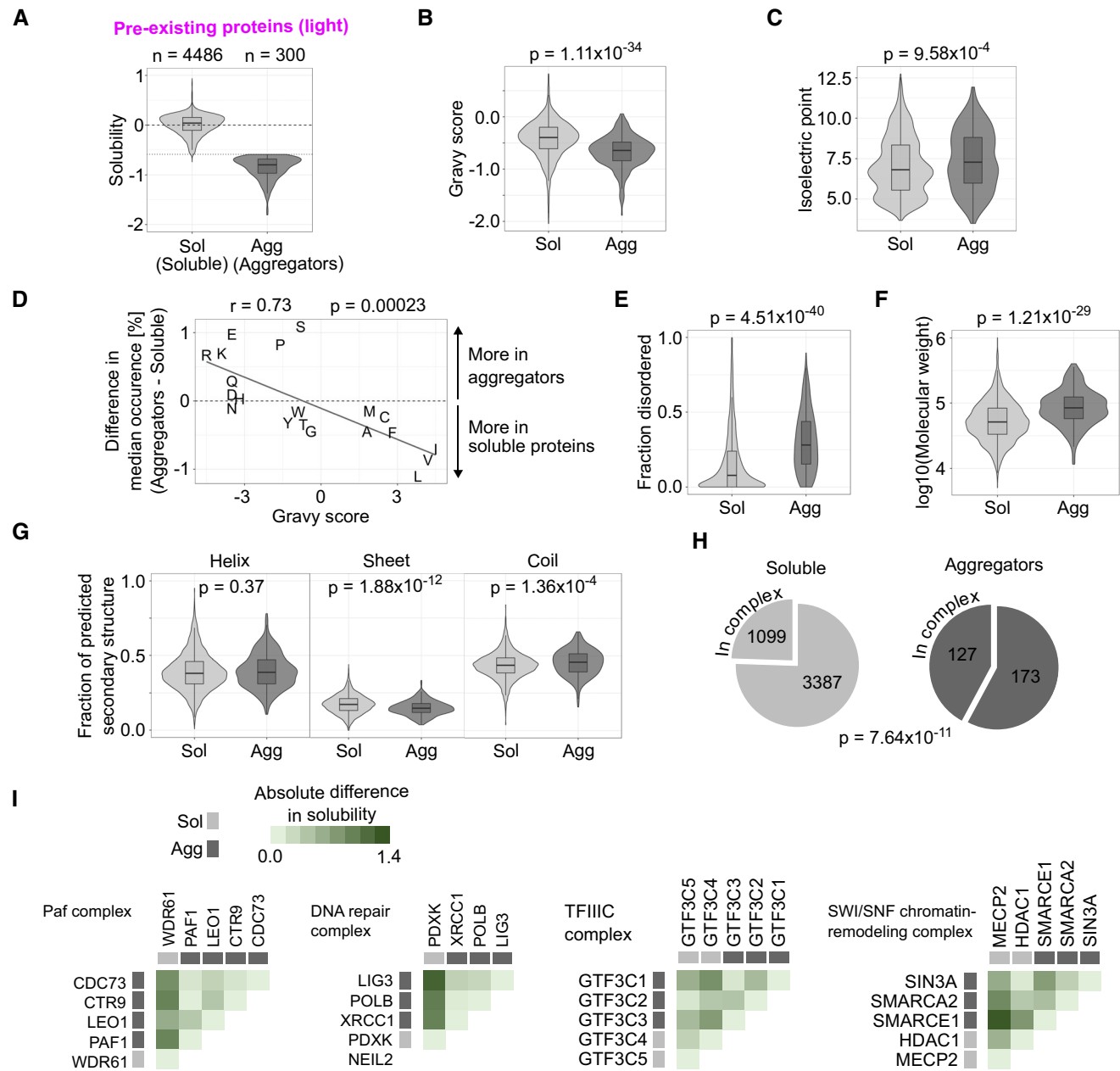

Figure 2.

**Figure 2. Characterization of proteins that aggregate in heat shock.**

A Definition of aggregating proteins. Solubility is measured as the $\log_2$-transformed ratio of protein intensities in the soluble fraction between heat-shocked and mock-shocked samples. Proteins with significant reduction in solubility [Benjamini–Hochberg-adjusted *P*-value < 0.05 and solubility < $\log_2(2/3)$] are then considered as aggregators. Dotted horizontal line shows the cutoff at solubility of $\log_2(2/3)$.

B–D Comparisons of physicochemical properties between soluble proteins and aggregators. Hydrophobicity (gravy scores) (B) and isoelectric points (C) are shown as combined violin and boxplots (*P*-values are for non-parametric Wilcoxon test). Difference in median amino acid composition between soluble proteins and aggregators is compared with hydrophobicity (gravy score) for each amino acid (D). In (D), Pearson coefficient (*r*) with *P*-value is shown for the correlation analysis.

E–G Comparison of structural features between soluble proteins and aggregators. Fraction of protein sequence predicted to contain intrinsically disordered regions (E), molecular weight (F), and fraction of protein sequence predicted to contain secondary structure elements (alpha helix, beta sheet, or coil) (G) are compared. *P*-values are for non-parametric Wilcoxon test.

H Protein complex members in soluble proteins and aggregators. Pie charts show the fraction of proteins annotated to be a member of a protein complex. The number of proteins in each segment is indicated. *P*-value is for Fisher's exact test.

I Protein complexes involving aggregators. Heatmaps show the absolute difference in solubility change after heat shock between each complex member. Protein complexes with at least 75% of its members quantified and containing at least 60% of its members as aggregators are shown. "DNA repair complex" = "DNA repair complex NEIL2-PNK-Pol(beta)-LigIII(alpha)-XRCC1".

Data information: See "Materials and Methods" for more detailed description of protein annotations used in (B–I). Boxplots indicate median, first, and third quartiles with whiskers extended to 1.5 times the interquartile range out from each quartile. Violin plots show the data distribution. Data shown for pre-existing protein fraction (light) quantified with at least two unique peptides from at least two biological replicates.

Source data are available online for this figure.

---

could stem from technical experimental differences or biological dissimilarities in the core structures between human and yeast stress granules (Jain *et al*, 2016).

Many features enriched in aggregators (such as the high amount of disordered regions) could be a result of them being also enriched in nuclear or DNA-binding protein. When analyzing the features presented in Fig 2 for similar enrichment or depletion as observed for aggregators, nuclear and DNA-binding proteins are enriched or depleted in the same features (Appendix Fig S3). However, the molecular weight was similar between nuclear and cytosolic proteins. In addition, predicted alpha helical content was significantly lower for DNA-binding and nuclear proteins. These results suggest that within nuclear proteins, large proteins with lower alpha helical content tend to aggregate.

Chromosome duplications can lead to gene overexpression. It has been shown that the protein overproduction is counteracted by aggregation (Brennan *et al*, 2019). As K562 cells contain aneuploidic chromosomes, we analyzed the frequency of aggregators in each chromosome to test whether aggregators would be over-represented in duplicated chromosomes. We found no difference in the occurrence of aggregators or soluble proteins in any of the 23 chromosome (Appendix Table S1) suggesting that protein overproduction does not contribute to aggregation propensity upon heat shock.

To analyze whether aggregators would have stronger preference for chaperones, we searched for Hsp70 binding motifs in them. A Hsp70-binding motif has been reported to contain four or five hydrophobic residues flanked by positively charged residues (Rüdiger *et al*, 1997). We found a slightly higher occurrence of the binding motifs in aggregators but the difference was not significant (*P* = 0.09582, Appendix Fig S4A). In addition, we could not find differences in the amino acid composition within the binding motif of aggregators as compared to the soluble proteins (Appendix Fig S4B).

Supersaturation (high concentration relative to solubility) has been shown to correlate with aggregation propensity (Ciryam *et al*, 2013, 2015). We found that supersaturation scores (Ciryam *et al*, 2013) were not higher for aggregators than for soluble proteins (Appendix Fig S5A). The supersaturation score has two components: protein concentration and structurally corrected aggregation propensity score (based on Zyggregator method (Tartaglia *et al*, 2008); Ciryam *et al*, 2013). While the supersaturation score was not higher for aggregators, we found a higher Zyggregator-based aggregation propensity score for them (Appendix Fig S5B). This suggests that the aggregation propensity in heat shock is more related to structural features of proteins rather than to supersaturation.

In summary, we observed heat shock-induced aggregation of nuclear, hydrophilic proteins with high molecular weight and intrinsically disordered regions. In addition, proteins that aggregated were more likely to be part of protein complexes.

## Disaggregation of heat-induced protein aggregates during recovery from heat shock

To monitor protein solubility during recovery, we measured protein intensities of the pre-existing proteins (light) in the soluble (NP-40 extractable) fraction. We sampled multiple time points and had a time-matched mock-shocked reference for each one of them (Fig 1A). Therefore, this approach allowed for fine-controlled measure of the solubility during recovery with high temporal resolution. The ratios between heat-shocked and mock-shocked samples were calculated at each time point and the $\log_2$-transformed ratios for pre-existing proteins (light) are shown in Fig 3A.

Proteins that stayed soluble after heat shock remained largely soluble during the recovery period (Fig 3A). However, most aggregators regained solubility during recovery from heat shock (Fig 3A). To quantitatively analyze dynamics of disaggregation, a linear model was fitted for each protein and the slope was used as an estimate for the disaggregation rate. The distributions of slopes (Fig 3B) indicate the steady solubility maintained with proteins that stay soluble after heat shock. In addition, the disaggregation of aggregators is evident from a positive shift of the slope values (Fig 3B). Similar observations were made with yeast (Wallace *et al*, 2015). Together, these results indicate that the disaggregation is the main strategy to deal with aggregates.

The total protein intensity remained constant during the recovery (Fig EV4A). This indicated that the increased intensity in the soluble fraction during recovery for aggregators stemmed from increasing

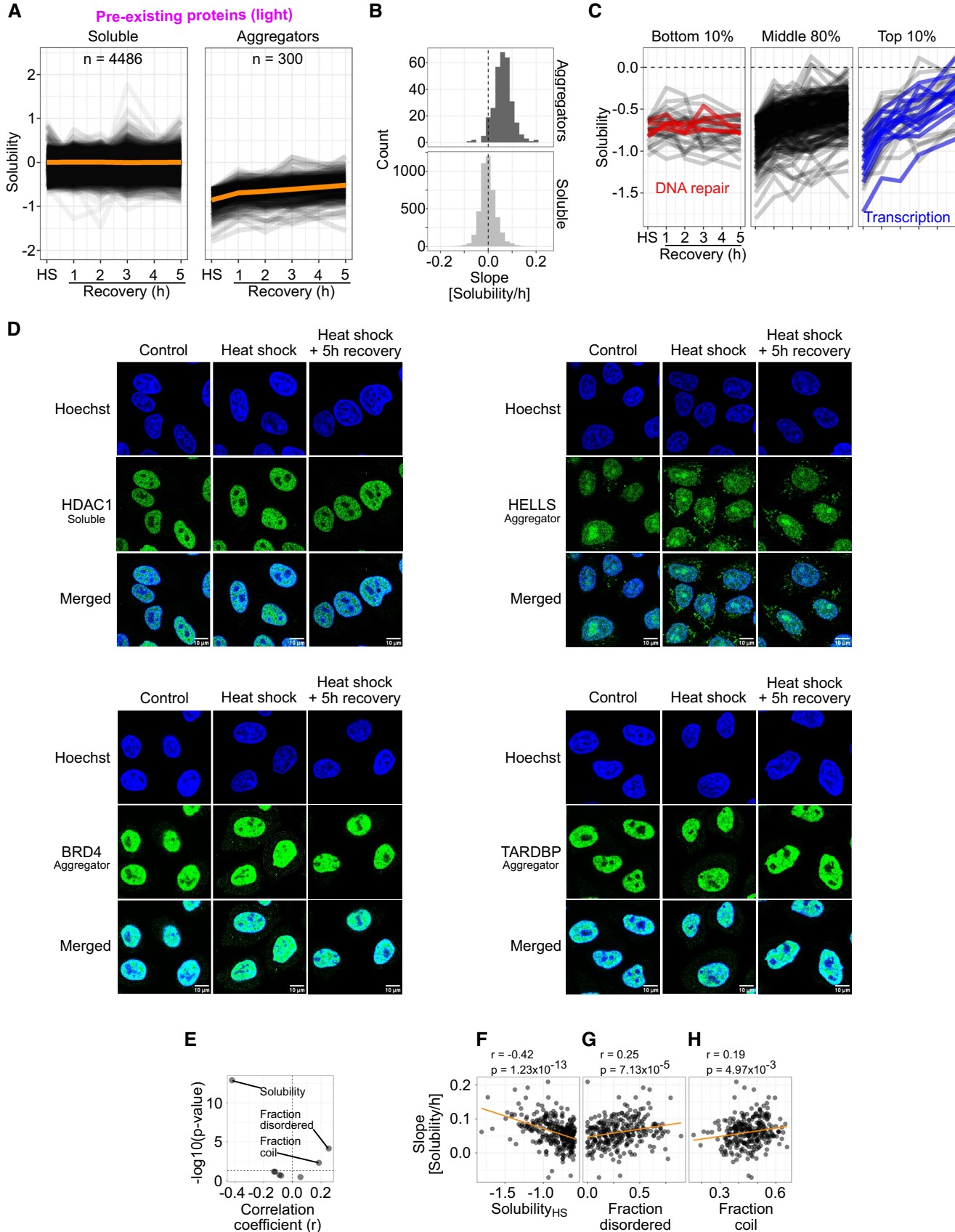

**Figure 3.**

**Figure 3.  Disaggregation of aggregated proteins during recovery from heat shock.**

A       Line graphs showing solubility after heat shock (HS) and during different time points of recovery. Each line corresponds to one protein. Orange lines show the mean solubility.

B       Quantification of disaggregation rates. Disaggregation rate for each protein is estimated as a slope from linear fits of data used in (A). Histograms of slopes (binwidth = 0.02 Solubility/h) are shown for aggregators and soluble proteins.

C       Different disaggregation profiles for aggregating proteins. Solubility line graphs as in (A) shown for aggregators with the bottom 10%, middle 80%, and top 10% of disaggregation rates. Proteins with bottom 10% of disaggregation rates and related to DNA repair are highlighted with red. Proteins with top 10% disaggregation rates and directly related to transcription are highlighted with blue.

D       Protein localization upon heat shock and during recovery as analyzed by immunofluorescence microscopy. HeLa cells were fixed and immunostained with antibodies against indicated target proteins (green) at different conditions [control, after heat shock (10 min at 44°C) and after 5 h of recovery from the heat shock]. DNA staining (Hoechst) is shown in blue.

E–H    Comparison of disaggregation rates with different protein characteristics enriched in aggregators (presented in Fig 2B, C, and E–G). Volcano plot presenting correlation coefficients and Benjamini–Hochberg-adjusted *P*-values (E; horizontal dashed line shows a *P*-value of 0.05). Scatterplot comparing disaggregation slopes and solubility change after heat shock (F), fraction of intrinsically disordered regions (G), and fraction of (random) coil-like secondary structure (H). In (F–H), scatterplots are shown for correlations with a *P*-value lower than 0.05. Correlation coefficients (*r*) with *P*-values are shown for Spearman's rank-order correlation.

Data shown for pre-existing protein fraction (light) quantified with at least two unique peptides from at least two biological replicates.

Source data are available online for this figure.

solubility. In addition, this also showed that heat shock did not induce protein degradation.

We took a more detailed look at the disaggregation patterns by concentrating on aggregators with the top or bottom deciles of slope values. Examination of aggregators with the lowest 10% of slope values revealed a small subset that were not disaggregated within 5 h of recovery (Fig 3C). These included proteins related to DNA damage: TDP1, FANCI, POLE, RIF1, and TIMELESS. Almost half of the aggregators with the highest 10% of slope values were transcription factors (FOXK2, MGA, ARID3A) or proteins closely related to transcription (TAF4, TCEB3, ELL, TRIM24, PRAME, BRD4, SMARCD2, SMARCE1, DAXX, and SCML2).

We validated the aggregation and disaggregation propensities of a few proteins from our dataset using immunofluorescence. HDAC1, a non-aggregating protein, showed nuclear localization in control conditions and remained unchanged upon heat shock and during recovery (Fig 3D). On the other hand, aggregators HELLS, BRD4, and TARDBP (all localized in the nucleus) showed increased intensity in the cytoplasm upon heat shock (Fig 3D). The increased cytoplasmic signal was strongest for HELLS (the most aggregating protein in the mass spectrometry experiment) while the cytoplasmic signal was weaker for BRD4 and TARDBP. HELLS seemed to form foci in the cytoplasm. Interestingly, prolonged heat shock caused the cytoplasmic and nuclear HELLS to localize and form foci at nuclear membranes (Appendix Fig S6). The cytoplasmic BRD4 formed bigger foci during prolonged heat shock while TARDBP formed nuclear foci or the intensity from non-foci proteins decreased (Appendix Fig S6).

After 5 h of recovery, a reduction in the cytoplasmic signal of BRD4 and TARDBP was observed, while the cytoplasmic HELLS remained in foci (Fig 3D). These findings corroborated the observations from the mass spectrometry experiment, where BRD4 and TARDBP disaggregated while HELLS remained aggregated in the insoluble fraction during the recovery. The solubility changes of nuclear proteins observed with mass spectrometry upon heat shock and during recovery coincide with protein transport to cytoplasm and foci formation.

Since aggregators were enriched in certain molecular features (Fig 2B–G), we wondered if these features would also be related to the rate of disaggregation. By analyzing the correlations between disaggregation slope and each of the features (Fig 3E), we found a significant (Benjamini–Hochberg-adjusted *P*-value < 0.05) negative

correlation between disaggregation slope and the loss of solubility after heat shock (Fig 3F; $P = 1.23 \times 10^{-13}$; $r = -0.42$). The proportion of disordered regions (Fig 3G; $P = 7.13 \times 10^{-5}$; $r = 0.25$) and fraction of (random) coil-like secondary structure (Fig 3H; $P = 4.97 \times 10^{-3}$; $r = 0.19$) had a weak but significant correlation with the disaggregation slope. The disaggregation rates were independent on whether or not the aggregators had an insoluble subpopulation at physiological conditions (Fig EV4B).

The correlation between solubility and disaggregation slopes could be a result of ratio compression during TMT quantification, which may stem from peptide co-fragmentation (Savitski *et al*, 2013). We therefore analyzed signal to interference values (which are lower for proteins prone for ratio compression) in the context of aggregation and disaggregation. We could not find any signs of lowered signal to interference values for aggregators (Appendix Fig S7A). In addition, no correlation was observed between signal to interference and solubility (Appendix Fig S7B) nor disaggregation slope (Appendix Fig S7C). Therefore, it is unlikely that ratio compression played a role in the observed correlation between solubility and disaggregation.

As mentioned earlier, aggregators were enriched in protein complex members (Fig 2H). Next, we explored their disaggregation as protein complex members in the recovery period. Within a protein complex ($n = 32$), aggregators had a weak trend for more similar disaggregation profiles when compared to scrambled complexes ($n = 10,000$) containing the same aggregators randomly re-distributed (Fig EV4C; $P = 0.048$). However, this coupling within complexes is not evident for the initial loss of solubility after heat shock (Fig EV2G; $P = 0.43$) suggesting that aggregators in complexes aggregate to different extent but can disaggregate similarly. However, as discussed earlier, complexes with a majority of aggregating members did aggregate coherently (Fig 2I).

To conclude, aggregated proteins were disaggregated during recovery from heat shock. The disaggregation rates dependent mainly on how much a protein aggregated in heat shock. In addition, larger extent of intrinsically disordered regions in a protein associated with faster disaggregation.

**Reversible stall in protein synthesis after heat shock**

The dynamic SILAC approach allowed to analyze also newly synthesized proteins (Fig 1A). To monitor protein synthesis, we quantified

newly synthesized proteins (heavy) from soluble fractions and compared each time point after heat treatment to a control collected before heat shock (Fig 1A). This approach allowed us to follow the accumulation of heavy-labeled proteins during recovery.

After mock shock, a steady rate of protein synthesis was observed (Fig 4A). The apparently fast synthesis rate—protein amount approximately doubled in 5 h—most probably stemmed from low starting amounts of heavy-labeled proteins: a small increase in the absolute protein amount will result in a large increase in the relative amount.

After heat shock, the accumulation of newly synthesized proteins slowed down globally (Figs 4B and C, and EV4D), in line with previous studies (Holcik & Sonenberg, 2005; Kirstein-Miles et al, 2013; Shalgi et al, 2013). However, during recovery the synthesis rates slowly increased and approached approximately the mock shock levels at the late time points of the recovery with exception of few proteins (Fig 4A and B).

The early medium switch in dynamic SILAC (90 min before heat shock) allowed incorporation of some heavy-labeled amino acids to newly synthesized proteins before heat treatment. Therefore, we could observe aggregation of newly synthesized proteins (Fig 4B and D). Aggregators identified from the pre-existing fraction (light) were predominantly the same proteins that aggregated in the newly synthesized fraction (heavy) (Fig 4D, Appendix Fig S8A). In addition, the solubility change upon heat shock was very similar for aggregators in both SILAC fractions (Appendix Fig S8A). The small differences between both protein populations did not correlate with disordered regions (Appendix Fig S8B).

At the end of the recovery period, few proteins showed a sharp increase in the newly synthesized fraction (Fig 4B and E). To analyze the upregulation of their synthesis, we looked at the protein intensities at the last time point, where the effect is most evident. The 10 most upregulated proteins included many heat shock proteins (Hsp): HSPA1B-HSPA1A (Hsp70), DNAJB1 (Hsp40), DNAJB4 (Hsp40), HSPB1 (Hsp27), HSPH1 (Hsp105), and SERPINH1 (Hsp47). Since HSPA1A and HSPA1B share high sequence similarity, we could not distinguish between the two paralogs in the mass spectrometry analysis and the results reflect a combination of the two. Interestingly, both highly upregulated Hsp40s (DNAJB1 and DNAJB4) belonged to class B of Hsp40s which are involved in protein disaggregation (Nillegoda et al, 2015). Although other Hsp40s were also upregulated (Appendix Fig S9), this suggested that the response to heat shock involved an increase in disaggregation capacity by upregulated protein synthesis.

Since some proteins that aggregated in heat hock were not disaggregated (Fig 3C), they possibly were replaced by upregulated protein synthesis. We could not find correlation between disaggregation and protein synthesis (Appendix Fig S10). However, it should be noted that the sample size in the analysis is relatively low due to the lower coverage of the newly synthesized protein fraction.

Next, we analyzed how the regulation of protein synthesis would match to transcriptional regulation after heat shock. We compared our results with previously reported changes of mRNA levels after heat shock (30 min at 42°C) in K562 cells (Vihervaara et al, 2017). Upregulation on mRNA level matched with upregulation on protein level only with the most upregulated transcripts (Fig 4F). From our 10 most upregulated proteins, majority of them were also upregulated on mRNA level (Fig 4F). We noticed that the upregulated heat shock proteins were the only ones with upregulation also on mRNA level; non-heat shock proteins were upregulated only on the protein level, suggesting that their upregulation was not driven by mRNA levels. Similar findings have been made with yeast (Muhlhofer et al, 2019).

To summarize, heat shock stalled translation. However, as the heat stress was removed the translation rates recovered accompanied by protein level upregulation including many heat shock proteins.

## Heat shock-induced changes in thermal stability

Next, we moved our focus to proteins that remain soluble after heat shock. We analyzed immediate heat shock-induced responses by applying two-dimensional thermal proteome profiling (2D-TPP) (Savitski et al, 2014; Becher et al, 2016; Mateus et al, 2017; Fig 5A). With this technique, we measured the propensity of heat-induced denaturation of soluble proteins after heat shock. In brief, after heat treatment, aliquots of cells were exposed to a short temperature gradient (3 min) that went well beyond the heat shock temperature (up to 66.3°C) eventually denaturing and aggregating all proteins in the sample (Fig 5A). Proteins from the soluble fraction were quantified and differences between heat shock and mock shock above 44°C (the heat shock temperature) indicated a heat shock-induced change in thermal stability (Fig 5A). We collected high-quality data (quantified with at least two unique peptides from minimum of six different temperatures) for 5,319 proteins with high reproducibility (Appendix Fig S11).

Strikingly, a large fraction of thermally stabilized proteins corresponded to aggregators (Fig 5B) including almost 90% of aggregators that were quantified in the assay (210 out of 234). Since the measurement was performed only on proteins that were soluble after heat treatment (mock shock or heat shock), the thermally stabilized proteins reflected the soluble remnants of aggregators. In other words, a sub-population of aggregators remained soluble after heat shock and required higher temperatures to aggregate.

Among the soluble remnants of aggregators, the strongest thermal stabilization was observed for ZC3H18 and ZMYM3, both proteins containing a zinc-finger domain (Fig 5B). Another zinc-finger containing protein (ZMYM4) was among the most thermally stabilized proteins.

When exploring the changes in thermal stability of proteins that were not aggregators, we observed some of the strongest effects for two mitogen-activated protein kinases (MAPKs) MAPK8 and MAPK9 (Fig 5B). They both belong to the c-Jun N-terminal kinases (JNK) group of MAP kinases, a branch in stress-activated MAPK pathway (Fig 5C) (Davis, 2000; Hotamisligil & Davis, 2016). We also observed changes in thermal stability of upstream kinases that are specific for JNKs (Fig 5C). We did not observe changes in thermal stability for p38 branch of stress-activated MAP kinases nor their specific upstream kinases (Fig 5C), although only one protein from each kinase level could be quantified. These results suggested that the activation of these pathways led to pronounced stability changes of proteins involved in them.

We detected thermal destabilization of RNA polymerase II subunits (Fig EV5) which was recently linked to detachment from DNA (Becher et al, 2018). This observation would be in line with the

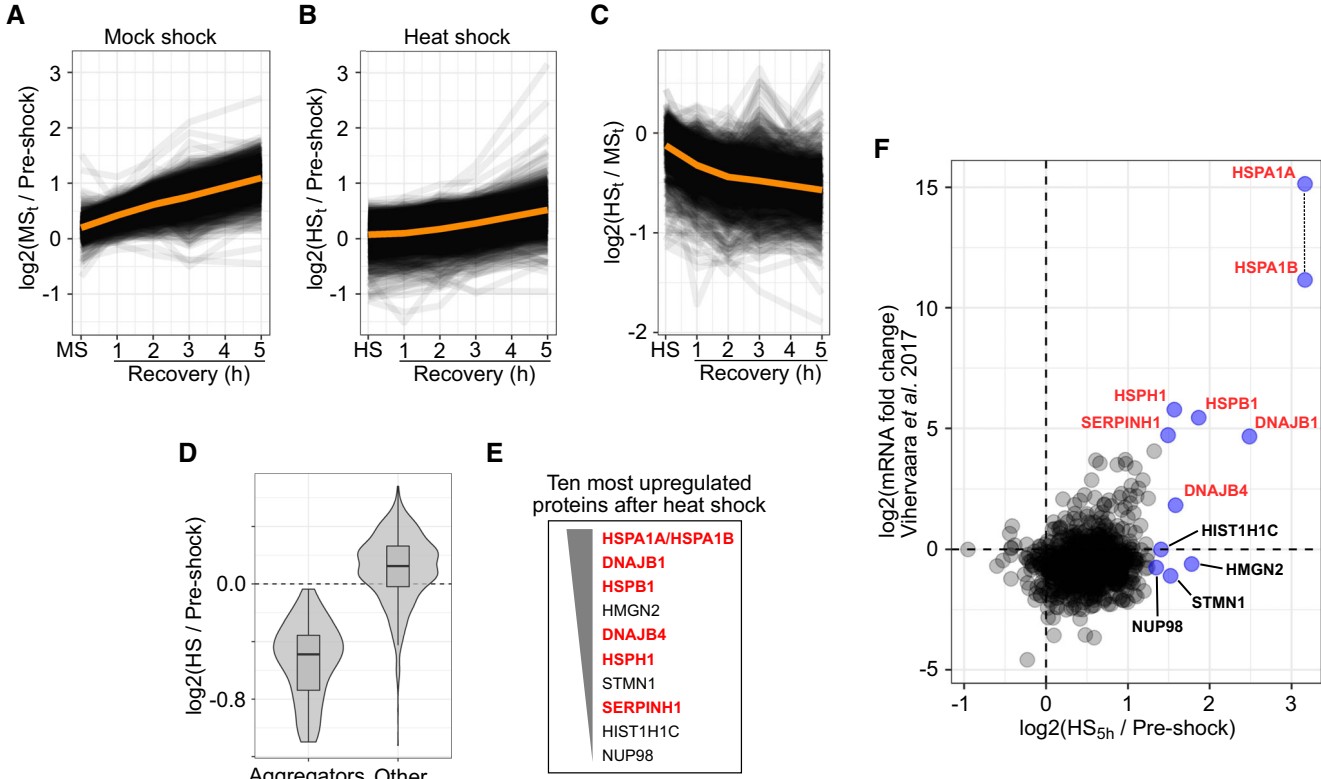

**Newly synthesized proteins (heavy)**

**Figure 4. Newly synthesized proteins after heat shock.**

A, B  Accumulation of newly synthesized proteins after mock shock (A) and heat shock (B). Line graphs showing the accumulation of signal intensity ($\log_2$-transformed ratios to pre-shock control) in the soluble fraction. Orange lines show the mean ratio.

C  Amount of newly synthesized proteins after heat shock compared with mock shock. Line graphs show the heat shock to mock shock ratio during recovery. Orange line shows the mean ratio.

D  Combined violin and boxplots show the $\log_2$-transformed ratio to pre-shock of newly synthesized proteins in the soluble fraction after heat shock. Proteins that aggregated in the pre-existing (light) fraction are compared with all other proteins. Boxplots indicate median, first, and third quartiles with whiskers extended to 1.5 times the interquartile range out from each quartile. Violin plots show the data distribution. Data from at least two biological replicates.

E  Ten proteins with the highest intensity at 5 h after heat shock (i.e., strongest upregulation) are listed in order of decreasing intensity. Heat shock proteins are highlighted with red.

F  Comparison of heat shock-induced protein synthesis and mRNA synthesis after heat shock (Vihervaara *et al*, 2017). Ten most upregulated proteins in the proteomics analysis are labeled and highlighted in blue; heat shock proteins are labeled red. HSPA1A and HSPA1B could not be distinguished from one another in the proteomics analysis and are plotted as they would have the same intensity (indicated with a vertical dashed line).

Data information: All proteomics data shown in (A–D and F) are quantified with at least two unique peptides from at least two biological replicates. Proteomics data shown for newly synthesized proteins (heavy). HS = heat shock. MS = mock shock.

Source data are available online for this figure.

global down-regulation of transcription upon heat shock (Vihervaara *et al*, 2017).

Some of the strongest thermal destabilizations were measured for H1 histones (Fig EV5), proteins that link nucleosomes together in compacted chromatin (Hergeth & Schneider, 2015). C-terminus of H1 histones is largely unstructured but it folds when bound to DNA (Roque *et al*, 2005). We reasoned that the thermal destabilization of H1 histones would correspond to partial unfolding and detachment from DNA upon heat shock—possibly resulting in opening of compact chromatin. In yeast, the human H1 histone homolog Hho1p was indeed reported to detach from repressed DNA upon heat shock (Zanton & Pugh, 2006).

Protein complex members showed thermal stability patterns that were possibly related to complex (de)activation or (dis)assembly as exemplified with 26S proteasome and CCT chaperonin complex (Fig 5D), both essential components of protein quality control. The thermal stability of both complexes decreased in heat shock. With the 26S proteasome the thermal destabilization was stronger for 19S regulatory particle than for the 20S core particle (Fig 5D). Interestingly, proteins from the 19S regulatory particle were thermally stabilized when ATP was added to cell lysates (Sridharan *et al*, 2019). However, the ATP levels were not altered during the heat shock (Fig EV1: viability measurements based on ATP quantification). The thermal destabilization of 19S regulatory particle could be linked to

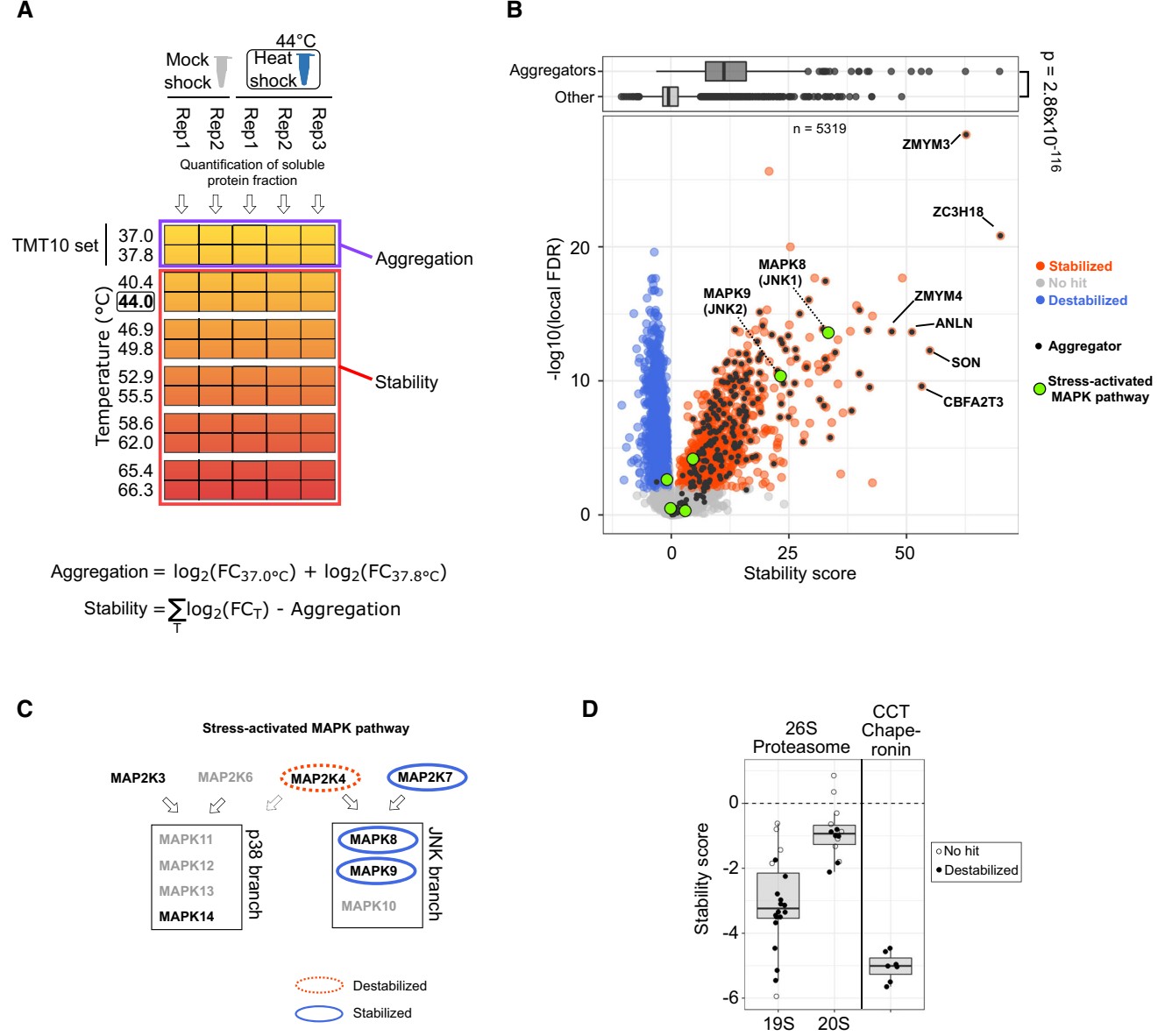

**Figure 5. Heat shock-induced changes in protein thermal stability.**

A  Experimental design to measure changes in thermal stability. Samples treated with either heat shock or mock shock are aliquot for treatment with temperatures that denature and aggregate all proteins. Samples are lysed with mild detergent (NP-40) and after tryptic digestion peptides from soluble protein fraction are labeled with TMT labels. The labeling is conducted so that samples from two adjacent temperature treatments are pooled and each TMT set is analyzed by mass spectrometer. At each temperature, the amount of protein still soluble is quantified and the fold change (FC) between heat shock and mock shock is calculated (see "Materials and Methods" for details). Fold changes in each temperature are summed after adjustment for heat shock-induced aggregation. Fold change-based thermal stability measures are finally transformed to scores for thermal stability (see "Materials and Methods" for details).

B  Volcano plot for the score for thermal stability. Significant (see "Materials and Methods") heat shock-induced stabilization and destabilization are indicated with blue and red, respectively. Proteins that aggregate in heat shock (as defined in Fig 2A) are shown in black—aggregators with the strongest stabilization are annotated. Members of stress-activated MAPK pathway highlighted in green—alternative names for MAP kinases are shown in parenthesis. Boxplot on top compares stability scores of aggregators (as defined in Fig 2A and other proteins). The P-value is for non-parametric Wilcoxon test.

C  Schematic for the structure of stress-activated MAPK pathway (adapted from Hotamisligil & Davis, 2016 and Davis, 2000) with the two branches of MAP kinases (p38 and JNK) and their upstream kinases (MKK3, MKK4, MKK6, and MKK7). Arrows show the target MAPK branch for each MAPKK. The role of MAP2K4 in p38 phosphorylation is unclear (Hotamisligil & Davis, 2016) and is presented with dashed line. Changes in thermal stability after heat shock are marked with ovals.

D  Thermal stability changes in protein complex members. Scores for thermal stability shown for proteins from 26S proteasome (separately for 19S regulatory and 20S core particles) and CCT chaperonin complex. Boxplots indicate median, first, and third quartiles with whiskers extended to 1.5 times the interquartile range out from each quartile. Data used in stability score calculation is from two (mock shock) and three (heat shock) biological replicates.

Data information: MAPK = mitogen-activated protein kinase, MAPKK = MAP kinase.
Source data are available online for this figure.

an impairment of ATP-driven proteasome activation observed in acute heat shock (Kuckelkorn *et al*, 2000). However, the inhibiting effect might be only a temporary response, since prolonged exposure to repeated heat shocks increases proteasomal activity (Beedholm *et al*, 2004).

Overall, we found a sub-pool of aggregation-prone proteins that resisted aggregation in heat shock. In addition, changes in thermal stability were observed in different cellular processes, such as stress signaling, DNA binding and with complexes related to protein quality control.

## Discussion

We developed a platform to study protein aggregation and disaggregation in human cells *in situ* after non-lethal heat shock. We found that heat shock induced the aggregation of proteins enriched in nuclear localization, intrinsically disordered regions, high molecular weight, and hydrophilic character.

The nuclear localization of aggregators could be linked to large extents of disordered regions found in the underlying protein, particularly since DNA-binding proteins are known to contain disordered regions (Fuxreiter *et al*, 2011; Vuzman & Levy, 2012). Previous studies have shown that components of Hsp70 chaperone system localize to nuclear organelles, such as nucleolus (Welch & Feramisco, 1984) and nuclear speckles (Deane & Brown, 2017) upon heat shock, suggesting for higher need of quality control measures at nuclear sites.

Previously, when mapping melting points of the human proteome, DNA-binding proteins were found to be the most unstable proteins (Savitski *et al*, 2014). Similarly, in bacteria, proteins with the lowest melting points include topoisomerases and proteins involved in DNA replication (Mateus *et al*, 2018). Therefore, unstable proteins might specifically relate to DNA.

Aggregation in the nucleus could mean several different things. For example, during stress, proteins have been reported to enter the nucleoli (Frottin *et al*, 2019) and increase abundance at chromatin (Aprile-Garcia *et al*, 2019). These could be linked to our results of increased thermal stability for soluble sub-pools of aggregators that did not aggregate in heat shock (Fig 5B); the most strongly thermally stabilized proteins were indeed zinc-finger containing DNA-binding proteins (Fig 5B).

We found that disordered regions are not only enriched in aggregators (Fig 2E), but the amount of disordered regions in a protein sequence correlated with the disaggregation rates (Fig 3G). Proteins containing disordered regions have been previously reported to be prone for aggregation (Uemura *et al*, 2018) and enriched in aggregates formed in *in vivo* models (Walther *et al*, 2015; Hosp *et al*, 2017). Walther *et al* (2015) propose that it could be a protective mechanism to actively sequester proteins with disordered regions, since these are often associated with aggregation diseases. In addition, the involvement of disordered regions in dosage sensitivity has been demonstrated (Vavouri *et al*, 2009; Bolognesi *et al*, 2016). Indeed, overexpressed disordered proteins from aneuploidic chromosomes are sequestered to aggregates (Brennan *et al*, 2019), degraded faster when present in complexes with super-stoichiometric amounts (McShane *et al*, 2016) and form toxic cytoplasmic granules when present with high concentration (Bolognesi *et al*,

2016). Therefore, aggregation of disordered proteins upon heat shock could protect cells from their potentially toxic effects.

Interestingly, the disordered regions *per se* are water soluble and might not contribute to the aggregation propensity of proteins (Uemura *et al*, 2018). Therefore, we speculate whether disordered regions could serve other functions not related to sequestering proteins to aggregates. The correlation between disordered regions and disaggregation rates could be explained by weaker intra-molecular interactions between proteins in aggregates. Disordered regions could also facilitate disaggregation by providing flexible loop regions for disaggregase(s) to act up upon. It has been proposed that, with amyloid fibers, Hsp70 disaggregase acts through flexible regions to exert pulling forces to the aggregated proteins (Wentink *et al*, 2019).

Enrichment of high molecular weight proteins in aggregates has been reported in different stress conditions, for example in yeast (Weids *et al*, 2016) and mice (Hosp *et al*, 2017). In addition, proteins were observed to lose more solubility when exposed to common precipitants if they had high molecular weight (Kramer *et al*, 2012). Together with our findings (Fig 2F), these results suggest that high molecular weight proteins are aggregation-prone and this is probably more due to their biophysical properties rather than to any biological reasons.

The formation of cytoplasmic foci that coincide with the aggregation detected in mass spectrometry experiment suggests that nuclear proteins do not aggregate in the nucleus, but rather translocate and aggregate in cytoplasm. It should be noted that the immunofluorescence detection cannot distinguish between aggregate formation and protein accumulation. In addition, the link between foci formation and loss of solubility is not trivial. Small aggregates that are beyond the detection limit of microscopy could contribute to a large fraction of the solubility decrease observed for aggregators (Mogk *et al*, 2018). In addition, majority of proteins related to cytoplasmic stress granules remained soluble (Fig EV3C). Therefore, we would hesitate to conclude that loss of solubility could be explained by cytoplasmic foci formation. Although some evidence connects foci formation and solubility change in mass spectrometry-based assays (Wallace *et al*, 2015), more focused studies would be required to investigate the issue.

We observed that the majority of aggregated proteins disaggregated during recovery (Fig 3A and B). These findings indicate that, as observed with yeast (Wallace *et al*, 2015), the main strategy for human cells to handle aggregates is disaggregation. However, it should be noted that proteasomal degradation, as well requires aggregate re-solubilization prior to degradation (Nillegoda *et al*, 2018). Therefore, it remains an open question whether the disaggregated proteins are destined for re-folding or degradation. Based on our data, at least in the 5 h following heat shock no degradation was observed (Fig EV4A).

Based on earlier work done with human disaggregase *in vitro* (Nillegoda *et al*, 2015) and in yeast (Wallace *et al*, 2015), one might expect a full disaggregation to take place in a time scale from minutes to approximately an hour. However, even after 5 h of recovery most of the aggregators are still on their way to being fully disaggregated (Fig 3A). Therefore, disaggregation in human cells seems much slower than in yeast. The reasons for this might stem from different heat shock conditions and how severe they are for human and yeast. Biologically, however, the simplest explanation

could be the Hsp100 disaggregation system present in yeast that can be more efficient than the human Hsp70-based system (Rampelt *et al*, 2012). Similar to humans, in *Caenorhabditis elegans* (another metazoan without the Hsp100-like disaggregase system), minute-scale disaggregation rates were observed for aggregated luciferase *in vitro* while traces of luciferase aggregates were found even days after heat shock *in vivo* (Kirstein *et al*, 2017). These results suggest that, although capable to disaggregate, metazoan disaggregase system is less efficient also in real cellular context than yeast Hsp100-based system.

We noted that the most upregulated heat shock proteins could, in theory, form a Hsp70-based disaggregase if assuming that the components would be co-expressed. These proteins include a Hsp70 (HSPA1A|HSPA1B) with a nucleotide exchange factor (HSPH1) and two Hsp40 proteins (DNAJB1 and DNAJB4). To speculate further, the two class B Hsp40s could reflect an adjustment toward clearance of larger aggregates, as it has been suggested for class B in contrast to class A (which are involved in clearing smaller size aggregates) (Nillegoda *et al*, 2015). However, it should be noted that other Hsp40s are upregulated too, for example DNAJC8, that has no known link to disaggregation. Further studies would help to understand the co-expression of disaggregase components and the possible adjustment of its composition to reflect the aggregation load.

The thermal stabilization of soluble remnants of aggregating proteins could reflect an instant post-translational mechanism of induced thermotolerance, although it should be noted that we cannot conclude how much of the increased stability signal is stemming from more stable protein sub-population that was already present in the sample before the heat shock and how much of the signal reflects heat shock-induced stability changes. For the heat shock-induced stabilization, the 2D-TPP assay, as we applied it here, can be viewed as a way to measure instantly gained thermotolerance without transcriptional or translational regulation. This is achieved by concentrating purely on protein solubility (direct measure of heat "sensitivity") and having no recovery time between the two heating steps (i.e., the heat/mock shock and the temperature gradient applied to aliquots). We speculate that the heat shock-induced stability could be done by a network of kinases, or other modifying enzymes, that modify proteins making them more stable. It would also be tempting to speculate that our results reflect the actions of chaperone networks [i.e., the epichaperome (Rodina *et al*, 2016)]. The limitation of the method is that it does not contain information about the reasons behind the stability changes. Therefore, it remains unsolved whether the stability changes are because of direct changes in proteins (e.g., post-translational modifications) or interactions with other molecules (e.g., chaperones or DNA). While the 2D-TPP assay discussed here reflects the response of mainly pre-existing proteins, it would be interesting to develop the method further to analyze heat-induced stability changes of newly synthesized proteins. For example, one could speculate that the stability of newly synthesized proteins could be affected by incomplete folding or chaperone binding. Extending the analysis by using chaperone inhibitors could help to understand these processes.

To conclude, we mapped protein solubility after non-lethal heat shock and during recovery *in situ* by hyperplexed proteomics. This allowed not only to characterize stress-induced aggregation but to study disaggregation patterns as well. Complementary, non-aggregating proteins were studied with 2D-TPP which allowed to explore solubility-independent heat shock-induced processes. While we introduced these techniques here to study heat-induced solubility changes, in the future it would be interesting to adapt the same approach to study and compare different stress condition that lead to protein aggregation such as chemical or other environmental stress conditions.

# Materials and Methods

### Reagents and Tools table

| Reagent/resource | Reference or source | Identifier or catalog number |
| --- | --- | --- |
| **Experimental models** | | |
| K-562 human chronic myelogenous leukemia cell line | ATCC | CCL-243 |
| HeLa cells | S. Narumiya | CVCL_1922 |
| **Antibodies** | | |
| Rabbit polyclonal anti-HELLS | Sigma | HPA063242-100UL |
| Mouse monoclonal anti-TARDBP | Novus Biological | NBP1-92695SS |
| Rabbit polyclonal anti-BRD4 | Sigma | HPA015055-100UL |
| Rabbit polyclonal anti-HDAC1 | Sigma | HPA029693 |
| Goat anti-rabbit AlexaFluor-488 | Thermo Fisher | A-11008 |
| Goat anti-mouse AlexaFluor-488 | Thermo Fisher | A-11001 |
| **Reagents** | | |
| L-lysine (heavy) 13C615N2 | Thermo Fisher Scientific | 88209 |
| L-arginine (heavy) 13C615N4 | Thermo Fisher Scientific | 89990 |

**Reagents and Tools table** (continued)

| Reagent/resource | Reference or source | Identifier or catalog number |
|---|---|---|
| cOmplete EDTA-free Protease inhibitor cocktail | Roche | 11873580001 |
| PhosSTOP (phosphatase inhibitor cocktail) | Roche | 4906845001 |
| NP-40 (IGEPAL CA-630) | Sigma-Aldrich | 18896 |
| Benzonase Nuclease | Millipore | 71206-3 |
| Hydroxylamine | Sigma-Aldrich | 438227 |
| TMT10 | Thermo Fisher Scientific | 90111 |
| TMT11 | Thermo Fisher Scientific | A37724 |
| Chloroacetamide | Sigma-Aldrich | C0267 |
| Tris(2-carboxyethyl)phosphine hydrochloride | Sigma-Aldrich | C4706 |
| Trypsin | Promega | V5111 |
| Lys-C | FUJIFILM Wako | 125-05061 |
| DMSO | Sigma-Aldrich | 276855 |
| 16% Paraformaldehyde | Thermo Fisher Scientific | 28908 |
| **Software** | | |
| isobarQuant | Franken *et al* (2015) | |
| | https://github.com/protcode/isob | |
| Mascot | Matrix Science | |
| | http://www.matrixscience.com/ | |
| R | R Core Team | |
| | https://www.R-project.org | |
| Fiji | ImageJ | |
| | https://imagej.net/Fiji/ | |
| **Databases** | | |
| UniProt | UniProt Consortium (2019) | |
| | https://www.uniprot.org/ | |
| Human Protein Atlas | Thul *et al* (2017) | |
| | http://www.proteinatlas.org | |
| Protein complexes | Ori *et al* (2016) | |
| | http://www.bork.embl.de/Docu/variable_complexes/ | |
| Database of Disordered Protein Predictions | Oates *et al* (2013) | |
| | http://d2p2.pro | |
| Gene Ontology Annotation Database | Huntley *et al* (2015) | |
| **Commercial kits and consumables** | | |
| CellTiter-Glo Luminescent Viability Assay | Promega | G7571 |
| Optiplate-96 Luminescence plate | PerkinElmer | |
| Filter plates (0.45 μm) | Merck Millipore | MSHVN4550 |
| Filter plates (0.22 μm) | Merck Millipore | MSHVN2250 |
| BCA Protein Assay Kit | Thermo Fisher Scientific | 23225 |
| Carboxylate modified magnetic particles, hydrophilic | Sigma-Aldrich | 45152105050250 |
| Carboxylate modified magnetic particles, hydrophobic | Sigma-Aldrich | 65152105050250 |
| Trapping cartridge. Acclaim PepMap 100 C18 LC column; 5 μm particles with 100 Å pores; 5 mm column with 300 μm inner diameter | Thermo Fisher Scientific | |
| Analytical column. nanoEase HSS C18 T3, 75 μm × 25 cm, 1.8 μm, 100 Å | Waters | |

Reagents and Tools table (continued)

| Reagent/resource | Reference or source | Identifier or catalog number |
|---|---|---|
| **Other** | | |
| UltiMate 3000 RSLC nano LC system | Thermo Fisher Scientific | |
| Q Exactive Plus Orbitrap mass spectrometer | Thermo Fisher Scientific | |
| Orbitrap Fusion Lumos mass spectrometer | Thermo Fisher Scientific | |
| SureCycler 8800 Thermal Cycler | Agilent | |
| Infinite M1000 PRO plate reader | TECAN | |
| 1290 Infinity (for HPLC fractionation) | Agilent | |

## Methods and Protocols

### Cell culture

K-562 cells (ATCC CCL-243) were cultured in SILAC RPMI 1640 medium (Thermo Fisher) supplemented with 2 mM L-gluta-mine, 0.96 mM L-lysine (light) (Thermo Fisher), 0.48 mM L-arginine (light) (Thermo Fisher), and 10% dialyzed FBS at +37°C (5% $CO_2$). For heavy SILAC medium, $^{13}C_6^{15}N_2$ L-lysine (Thermo Fisher) and $^{13}C_6^{15}N_4$ L-arginine (Thermo Fisher) were used keeping their molar concentration same as in light. The K562 cells were chosen based on their tolerance to a 45°C heat shock (Mivechi, 1989; Fig EV1) and easy handling (suspension cells).

### Heat shock and recovery with dynamic SILAC

- Experiment 1:
  ○ Analysis of soluble protein fraction after 0, 1, 2, 3, and 5 h of recovery from heat shock (10 min, 44°C)
  ○ Heat-shocked samples compared with time-matched mock controls
  ○ Samples lysed with weak non-ionic detergent (NP-40)

- Experiment 2:
  ○ Analysis of soluble and total protein fraction before and after 0, 1, 3, and 5 h from heat shock (10 min, 44°C)
    □ Soluble fraction: samples lysed with weak non-ionic detergent (NP-40)
    □ Total protein fraction: samples lysed with strong ionic deter-gent (SDS)

The protocol for both Experiments is as follows (differences between Experiment 1 and 2 are point out):

Medium switch:
- Transfer cells with fully adapted light amino acids to a conical centrifugation tube
- Approximately $2.5 \times 10^6$ cells needed for each replicate and Experiment
- Pellet cells [190 × g, 3 min, room temperature (RT)]
- Gently aspirate supernatant
- Re-suspend cells to 1 ml of pre-warmed (37°C) heavy SILAC medium
- Pellet cells (190 × g, 3 min, RT)
- Gently aspirate supernatant
- Re-suspend cells to pre-warmed (37°C) heavy SILAC medium to a final cell density of $5 \times 10^5$ cells/ml.

- Transfer cells to a cell culture flask and incubate 90 min at +37°C (5% $CO_2$)

Heat treatment:
- Prior to heat treatment, collect pre-shocked sample (see "Sample collection")
- Distribute cells to 96-well PCR plates (200 μl/well, $10^5$ cells/well)
  ○ Experiment 1: two plates (mock and heat shock) with five samples on each plate
  ○ Experiment 2: one plate (heat shock) with 10 samples on the plate
- Seal plates with aluminum foil
- Place plates on pre-warmed heat blocks for 10 min
  ○ Experiment 1 and 2: Heat shock at 44°C
  ○ Experiment 1: Mock shock at 37°C

Recovery from heat shock:
- After heat treatment, remove the aluminum foil
- Seal plates with vent filter membrane
- Incubate cells at 37°C (5% $CO_2$)

Sample collection:
- Collect samples right after heat shock and 1, 2, 3 and 5 h after recovery
  ○ Experiment 2: skip the sample collection at 2 h of recovery
- Remove vent filter membrane and transfer cells to 0.2 ml strip tubes
- Wash cells:
  ○ Pellet cells down (1,000 × g, 1 min, RT)
  ○ Remove 90% of the supernatant (i.e., 180 μl)
  ○ Re-suspend cells to 180 μl of ice cold PBS
  ○ Repeat the washing
  ○ Pellet cells down (1,000 × g, 1 min, RT)
  ○ Remove 90% of the supernatant (i.e., 180 μl)
  ○ Snap-freeze cells in liquid nitrogen
  ○ The samples can be stored in −80°C for later processing

Cell lysis:
- Thaw cells on ice
- add 30 μl of 5/3× concentrated lysis buffer to a final concentration of
  ○ 50 mM HEPES
  ○ 0.8% NP-40
    □ Experiment 2: to collect total protein fraction, use 1% SDS

- 1.5 mM MgCl$_2$
- 1× protease inhibitor cocktail
- 1× phosphatase inhibitor cocktail
- 0.25 U/µl benzonase
- pH ≈ 7.4
- Incubate at 4°C on a shaker for 1 h
  - Experiment 2: incubate total protein fractions 30 min at RT to avoid SDS precipitation

### Two-dimensional thermal proteome profiling

Preparing cells for the experiment:
- Transfer cells to a conical centrifugation tube
  - Approximately $1.5 \times 10^7$ cells needed for each replicate
- Pellet cells (190 × g, 3 min, RT)
- Gently aspirate supernatant
- Re-suspend cell to medium to a final cell density of $5 \times 10^5$ cells/ml.

Heat treatment:
- Distribute cells to two 96-well PCR plates (200 µl/well, $3 \times 10^5$ cells/well)
- Seal plates with aluminum foil
- Place plates on pre-warmed heat blocks for 10 min
  - Heat shock at 44°C
  - Mock shock at 37°C

Thermal shift assay:
- After heat treatment, pool mock- and heat-shocked samples to separate conical centrifugation tubes
- Pellet cells (180 × g, 3 min, RT)
- Wash cells:
  - Gently aspirate supernatant
  - Re-suspend cells to 30 ml of PBS (RT)
  - Pellet cells (180 × g, 3 min, RT)
  - Gently aspirate supernatant
- Re-suspend cells to PBS (RT) to a final cell density of $5.5 \times 10^6$ cells/ml
- Aliquot cells to a 96-well PCR plate (100 µl/well, $5.5 \times 10^5$ cells/well)
  - Mock- and heat-shocked samples each in separate rows (e.g., mock-shocked samples aliquot to row A and heat-shocked to row B)
- Pellet cells on the plate (390 × g, 2 min, RT)
- Gently aspirate 80% of the supernatant (80 µl)
- Carefully re-suspend pelleted cells to the remaining PBS
- Put the plate on a thermal cycler for 3 min
  - Temperature gradient: 37.0; 37.8; 40.4; 44.0; 46.9; 49.8; 52.9; 55.5; 58.6; 62.0; 65.4; and 66.3 °C
- Equilibrate samples for 3 min at RT before placing the plate on ice

Cell lysis:
- add 30 µl of 5/3× concentrated lysis buffer to a final concentration of
  - 50 mM HEPES
  - 0.8% NP-40)
  - 0.25 U/µl
  - 1.5 mM MgCl$_2$
  - 1× protease inhibitor cocktail
  - 1× phosphatase inhibitor cocktail
  - pH ≈ 7.4

- Incubate at 4°C on a shaker for 1 h
- Lysates can be stored in −80°C for later processing

### Extraction of soluble protein fraction
- Pre-wet 96-well filter plate (0.45 µm):
  - Add 50 µl of lysis buffer to each well
  - Place filter plate of top of a 96-well collection plate
  - Centrifuge lysis buffer through the filter (500 g, 5 min, 4°C)
  - Discard the collection plate
- Transfer lysates on to a pre-wet 96-well filter plate.
- Place filter plate of top of a 96-well collection plate
- Centrifuge lysates through the filter (500 g, 5 min, 4°C)
- The flow-through on the collection plate contains the soluble fraction
  - Optional: measure protein concentration (with, e.g., BCA assay) or dry samples on a vacuum concentrator
- Samples can be stored in −80°C for later processing

### Viability assay
Cell viability after heat shock and during recovery was estimated with CellTiter-Glo Luminescent Cell Viability Assay (Promega). Cells were pelleted, density was adjusted to $5 \times 10^4$ cells/ml with fresh medium and aliquot to 96-well PCR plates ($10^4$ cells/well). Cells were exposed to a heat treatment for 10 min in thermal cycler (Agilent SureCycler 8800) with different temperature for each aliquot (37.0; 39.6; 41.5; 43.6; 45.7; 47.5; 49.6; 52.0; 54.3, and 54.9°C).

After heat treatment, cells were allowed to recover at 37°C (5% CO$_2$). Samples were collected after heat shock and after 5 h of recovery. Half of each sample ($5 \times 10^3$ cells) was transferred to a luminescence plate (PerkinElmer Optiplate-96). Cells from the remaining half were pelleted, and equal volume (to match the sample) of supernatant was transferred to the luminescence plate (as blank measures). Equal volume of CellTiter-Glo reagent was added to each well, plate was mixed on a shaker for 2 min, and luminescence was recorded after 10 min. Luminescence was measured with TECAN Infinite M1000 PRO.

Blank measurement was subtracted from each sample. All samples were done as technical triplicates.

### Protein extraction and digestion
For protein extraction, we used a modified version of SP3 sample preparation protocol (Hughes et al, 2014; Moggridge et al, 2018; Smits et al, 2019). Samples (5–15 µg of protein depending on sample availability) were mixed with 47.6% ethanol/2.4% formic acid binding buffer containing carboxylate modified magnetic particles (Sera-Mag SpeedBead Carboxylate Modified Magnetic Particles, Hydrophilic Ref#45152105050250, Hydrophobic Ref#65152105050250). Proteins were let to bind to particles for 15 min at RT on shaker. Particle-bound proteins were transferred to a filter plate (Millipore MultiScreen, 0.22 µm, MSGVN2250) and centrifuged (1,000 g, 1 min, RT) to remove binding buffer. Proteins were washed four times with 70% ethanol and digested over-night in a digestion solution [90 mM HEPES, 5 mM CAA, 1.25 mM TCEP, 200 ng/sample trypsin (Promega, V5111), 200 ng/sample Lys-C (FUJIFILM Wako, 125-05061)] at RT on a shaker.

After digestion, peptides were collected by centrifugation (1,000 *g*, 1 min, RT). Residual particle-bound peptides were eluted with 2% DMSO, collected by centrifugation (1,000 g, 1 min, RT), and added to the original peptide sample. Samples were dried.

### TMT labeling

Peptides were dissolved in water and TMT labels (Thermo Fisher TMT10plex, TMT11-131C) (dissolved in acetonitrile) were added (with final acetonitrile concentration of 28.6%). Labeling reaction was conducted at RT for 1 h on a shaker. The reaction was quenched with 1.1% hydroxylamine for 15 min. Labeled samples were pooled and diluted with 0.05% formic acid to decrease acetonitrile concentration below 5%.

The labeling scheme for recovery assay with 11 TMT labels was as follows: mock-shocked samples (mock shock, 1, 2, 3, and 5 h of recovery), pre-shock control and heat-shocked samples (heat shock, 1, 2, 3, and 5 h of recovery) given in the order of increasing TMT reporter ion mass (i.e., from 126 to 131C) (see Fig 1B).

The labeling scheme for 2D-TPP with 10 TMT labels was as follows: mock shock replicate one (temperature one: TMT126, temperature two: TMT129N), mock shock replicate two (127N, 129C), heat shock replicate one (127C, 130N), heat shock replicate two (128N, 130C), and heat shock replicate three (128C, 131). In other words, samples from two adjacent temperatures of the temperature gradient were combined in each TMT set (see Fig 5A).

### Peptide desalting

Samples were transferred to an OASIS microplate (Waters HLB µElution plate, 186001828BA) for desalting. After binding peptides to the columns, they were washed two times with 0.05% formic acid and finally eluted with 0.05% formic acid/80% acetonitrile. Peptides were dried.

### Offline fractionation

Samples were dissolved in 20 mM ammonia and injected for reverse phase fractionation under high pH conditions. Samples were fractionated to 32 fraction and partially pooled to reduce the amount of fractions to 12. Fractions were dried.

### Quantitative mass spectrometry

Peptides were dissolved in 0.1% formic acid and subjected to liquid-chromatography using an UltiMate 3000 RSLC nano LC system (Thermo Fisher Scientific). The LC system was equipped with a trapping cartridge (Acclaim PepMap 100 C18 LC column: 5 µm particles with 100 Å pores, 5 mm column with 300 µm inner diameter) for online desalting and an analytical column (Waters nanoEase HSS C18 T3, 75 µm × 25 cm, 1.8 µm, 100 Å) for separation. Peptides were loaded on the trapping cartridge for 3 min with 0.05% TFA in LC-MS grade water at a flow rate of 30 µl/min. Peptides were eluted using buffers A (0.1% formic acid in LC-MS grade water) and B (0.1% formic acid in LC-MS grade acetonitrile) using increasing concentrations of buffer B at a flow rate of 0.3 µl/min. During a total analysis time of 120 min, the concentration of buffer B increased from initial 2–4% in the first 4 min, to 8% in the next 2 min, to 28% in the next 96 min, and finally to 40% in the next 10 min, followed by a 4 min washing step at 85% B before returning to initial conditions.

Peptides were injected to either a Q Exactive Plus Orbitrap (QE Plus, Thermo Fisher Scientific) or Orbitrap Fusion Lumos (FL) both using a Nanospray Flex ion source. In the following, the parameters are given for QE Plus and in parenthesis for FL. Mass spectrometers were operated in positive ion mode with spray voltage of 2.3 kV (2.4 kV) and capillary temperature of 275°C (300°C). Full scan MS spectra were acquired for a mass range of 375–1,200 $m/z$ (375–1,500 $m/z$) were in profile mode with a resolution of 70,000 (120,000) with a maximum fill time of 250 ms (64 ms) or automatic gain control with a maximum of $3 \times 10^6$ ions ($4 \times 10^5$ ions).

On the MS scan, data-dependent acquisition was applied by selectively fragmenting top 10 peptide peaks (3 s cycle time) with a charge state of 2–4 (2–7) using dynamic exclusion window of 30 s (60 s) and mass window of 0.7 $m/z$ (0.7 $m/z$) for isolation. Selected peptides were fragmented with normalized collision energy of 32 (38). MS/MS spectra were acquired in profile mode with a resolution of 35,000 (30,000) and an automatic gain control target of $2 \times 10^5$ ions ($1 \times 10^5$ ions). The first mass was set to 100 $m/z$.

### Data analysis

Raw mass spectrometry data were processed with isobarQuant (Franken *et al*, 2015). For protein identification (against human database in UniProt), Mascot search engine was used with the following search parameters: digestion with trypsin, maximum of three missed cleavages, 10 ppm peptide tolerance, and 0.02 Da MS/MS tolerance; carbamidomethylation of cysteines and TMT labels on lysine as fixed modifications; acetylation of N-terminus, methionine oxidation, and TMT label on N-terminus as variable modifications. For SILAC-TMT data, two separate Mascot searches were conducted: first with the settings described above for light and then a slightly modified search for heavy. The heavy search included a 10 Da heavier arginine and 8 Da heavier TMT label attached to lysine as fixed modifications. The rationale for using heavier TMT label on lysine was to mimic 8 Da heavier lysine since Mascot does not allow for two separate modifications for one amino acid—in this case a heavy lysine and a TMT tag.

After peptide and protein identification with Mascot, peptide level quantification (based on TMT reporter intensities) was conducted and peptide intensities were summed to protein level with isobarQuant (Franken *et al*, 2015). The isobarQuant output (protein level data) was imported to R (https://www.R-project.org). Proteins identified as contaminants or reverse database hits were filtered out. In addition, only proteins quantified with at least two unique peptides were kept for the following analysis. Protein intensities were then log$_2$-transformed.

Batch effects were removed from protein intensities of each TMT channel with R package *limma* (Ritchie *et al*, 2015) using *removeBatchEffect* function. Resulting intensities were normalized using variance stabilization (vsn) method with R package *vsn* (Huber *et al*, 2002). Missing values were imputed with R package *MSnbase* (Gatto & Lilley, 2012) using *impute* function.

For SILAC data in recovery assay, we used a normalization approach where protein intensities from light-labeled proteins (pre-existing proteins) were first normalized and these normalization coefficients were applied to heavy-labeled (newly synthesized) proteins. We justify this approach since light-labeled proteins generally should have consistent intensities throughout the time course while the intensities of heavy-labeled proteins should increase over

time. Therefore, these patterns are preserved throughout the analysis. In addition, it is expected that the intensity and coverage of heavy labeled proteins is relatively low and thus any normalization based on them would be strongly biased toward high abundant protein species. It is worth mentioning that any background degradation in pre-existing fraction (light) present in both, mock- and heat-shocked samples, is masked away by the normalization approach, although reported protein half-lives are generally much longer than 5 h (Schwanhausser *et al*, 2011; Mathieson *et al*, 2018).

Ratios between heat shock and mock shock were calculated for each time point. For heavy-labeled proteins, a ratio against pre-shocked control was calculated separately for heat-shocked and mock-shocked samples.

To statistically examine the solubility changes in heat shock, a LIMMA analysis was used to test for difference in heat shock/mock shock ratios (referred to as solubility in the main text) in light-labeled proteins. A difference was assigned significant if Benjamini–Hochberg-adjusted *P*-value was below 0.05 and fold change below 2/3. These proteins are referred to as aggregators in the main text.

2D-TPP data were analyzed as described before (Becher *et al*, 2018). Briefly, within all conditions, each temperature was normalized with vsn separately and ratio against 37°C sample in the temperature gradient was calculated for each temperature. The principle behind calculating scores for thermal stability is based on summing up differences between heat- and mock-shocked samples in every temperature point; to correct for the aggregation already taken place in heat-shocked sample, the average difference in the first two temperature points (37.0 and 37.8°C) were subtracted from all temperature points. When correcting for the aggregation, we assume that no aggregation has taken place in the mock-shocked sample in these temperatures. In practice, an iterative bootstrapping approach was used for each protein: data from one replicate were randomly selected for each temperature and scores for thermal stability were calculated within 500 rounds. These iterated scores were transformed to *z*-scores, and their mean was tested for deviation from zero (i.e., no change in thermal stability). From that comparison, Benjamini–Hochberg-adjusted *P*-value was calculated for each protein to estimate local false discovery rate (FDR). Finally, the means from every protein were transformed to *z*-scores and depict the final score for thermal stability. R package *fdrtool* (Strimmer, 2008) was used to calculate global FDRs. Proteins quantified from less than six temperatures were filtered out. Protein was assigned as hit if both, local and global FDR, were below 0.01; destabilized hits had a negative score for thermal stability and stabilized hits had a positive score for thermal stability.

Protein localization annotations were from Human Protein Atlas (www.proteinatlas.org) (Thul *et al*, 2017). Annotation with reliability levels of "Approved", "Supported" or "Validated" were included.

Gravy score for each protein was calculated as a sum of the values assigned to each amino acid in a protein sequence: arginine (−4.5), lysine (−3.9), asparagine (−3.5), aspartate (−3.5), glutamine (−3.5), glutamate (−3.5), histidine (−3.2), proline (−1.6), tyrosine (−1.3), tryptophan (−0.9), serine (−0.8), threonine (−0.7), glycine (−0.4), alanine (1.8), methionine (1.9), cysteine (2.5), phenylalanine (2.8), leucine (3.8), valine (4.2), and isoleucine (4.5).

Isoelectric points and molecular weights were calculated using R package Peptides (Osorio *et al*, 2015).

The predicted fraction of intrinsically disordered regions in proteins was from $D^2P^2$ database (Oates *et al*, 2013).

Protein secondary structure prediction was done with R package DECIPHER (Wright, 2016) using default settings. For each secondary structure element (sheet, helix, or coil), the predicted proportion in the protein sequence was calculated.

The amino acid sequence of the canonical isoform for each protein was used as input for calculating gravy score, isoelectric point, molecular weight, or predicting secondary structure elements.

For protein complex annotations, a manually curated database integrated from multiple sources (Ori *et al*, 2016) was used (including complexes with minimum of five distinct proteins).

Gene ontology (GO) term enrichments were conducted with R package clusterProfiler (Yu *et al*, 2012). Enrichment analysis was based on hypergeometric test with a cutoff of 0.05 for Benjamini–Hochberg-adjusted *P*-value.

List of human proteins linked to GO term "cytoplasmic stress granule" (GO:0010494) was collected from Gene Ontology Annotation Database (Huntley *et al*, 2015).

In comparisons between means of distributions (Figs 2B, C, E–G and 5C, and EV4) and correlation analysis (Figs 2D and 3E–H), the normality of distributions was estimated with a Shapiro–Wilk test. A distribution was assigned to be normally distributed if *P*-value in the test was at or above 0.05. In comparisons between two normally distributed data, a parametric test was used (*t*-test, Pearson correlation); otherwise a non-parametric alternative was used (Wilcoxon test, Spearman correlation). The used tests are indicated in the figure legends.

### Immunofluorescence microscopy

HeLa cells were grown in Dulbecco's minimal essential medium containing 2 mM L-Glutamine and 10% fetal calf serum at +37°C (5% $CO_2$). 20,000 cells in 200 µl of medium were seeded on each well of 8-well LabTek chambered imaging plates and incubated at +37°C with 5% $CO_2$ for 24 h. The media was changed 30 min before the start of the heat shock experiment. For heat shock, cells were transferred to an incubator set to +44°C (5% $CO_2$) for 10 min. For recovery, cells were transferred back to +37°C (5% $CO_2$). The control cells remained in +37°C (5% $CO_2$) throughout the experiment.

Cells were washed with PBS and fixed with 4% paraformaldehyde for 15 min at room temperature. Subsequently, the cells were permeabilized with 0.05% Triton X (10 min at room temperature) and washed twice with PBS. Non-specific binding was blocked with 3% BSA in PBST (0.1% Tween 20 in PBS) for 1 h at room temperature. Following primary antibody, dilutions were prepared in 3% BSA containing PBST; 1:1,000 anti-BRD4, 1:1,000 anti-HDAC1, 1:500 anti-HELLS, and 1:1,000 anti-TARDBP. After blocking, the cells were incubated with the diluted primary antibodies for 22–24 h at 4°C. Goat anti-mouse and Goat anti-rabbit cross-adsorbed secondary antibodies tagged with AlexaFluor-488 were diluted at 1:2,000 ratio in 3% BSA containing PBST. Hoechst 33342 was added to the diluted secondary antibody solution at a final concentration of 5 µg/ml. Cells were washed three times with PBST before incubating them with Hoechst-containing secondary antibodies. Following an incubation at room temperature for 90 min, the cells were washed three times with PBST and stored PBS for imaging.

Cells were imaged on Zeiss 780 NLO confocal microscope using a 63×/1.4 oil immersion objective and argon laser. The images were acquired with the following settings for the different fluorophores: Hoechst—Ex: 405 nm, Em: 410–479 nm and AlexaFluor-488—Ex: 488 nm, Em: 489–585 nm. The microscope was controlled using Zen 2012 software. The images were processed using Fiji, ImageJ.

## Data availability

The datasets produced in this study are available in the following database: Proteomics data: PRIDE PXD017291 (https://www.ebi.ac.uk/pride/archive/projects/PXD017291).

Expanded View for this article is available online.

## Acknowledgements
We thank members of the Savitski laboratory for discussions. We thank André Mateus for insightful and critical review of the manuscript. We thank the Proteomics Core Facility at the European Molecular Biology Laboratory (EMBL) for expert support. This work was supported by the EMBL. Open access funding enabled and organized by Projekt DEAL.

## Author contributions
MMS supervised the study. MMS, TAM, MR, and FS designed the study. TAM, MR, and DH performed the experiments. SS designed, performed, and analyzed immunofluorescence microscopy experiments. MMS, TAM, NK, and FS analyzed the data. MMS and TAM wrote the manuscript. All authors critically reviewed the manuscript.

## Conflict of interest
The authors declare that they have no conflict of interest.

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
