## [Review Process File · Molecular Systems Biology]

Aggregation and Disaggregation Features of the Human Proteome

Tomi Määttä, Mandy Rettel, Sindhuja Sridharan, Dominic Helm, Nils Kurzawa, Frank Stein, and Mikhail Savitski

DOI: [10.15252/msb.20209500](https://doi.org/10.15252/msb.20209500)

Corresponding author(s): Mikhail Savitski (mikhail.savitski@embl.de)

Review Timeline:

Submission Date:	5th Feb 20
Editorial Decision:	12th Mar 20
Revision Received:	16th Jul 20
Editorial Decision:	7th Aug 20
Revision Received:	25th Aug 20
Accepted:	9th Sep 20

Editor: Jingyi Hou

Transaction Report:

Thank you again for submitting your work to Molecular Systems Biology. We have now heard back from the three reviewers who agreed to evaluate your study. As you will see below, the reviewers acknowledge that the presented findings seem potentially interesting. They raise however a series of concerns, which we would ask you to address in a major revision.

I think that the reviewers' recommendations are rather clear and there is no need to repeat the comments listed below. Importantly, reviewer #3 mentioned that further experimental validations would be required to conclusively support the main findings. The reviewers also recommended extending the study by including further follow-up analyses in order to enhance the level of novel biological insight provided by the study. We think that addressing these concerns is important and would indeed significantly strengthen the manuscript.

All other issues raised by the reviewers need to be satisfactorily addressed as well. As you may already know, our editorial policy allows in principle a single round of major revision and it is therefore essential to provide responses to the reviewers' comments that are as complete as possible. Please feel free to contact me in case you would like to discuss in further detail any of the issues raised by the reviewers.

On a more editorial level, we would ask you to address the following issues.

REFEREE REPORTS

Reviewer #1:

Summary:

The manuscript by Määttä et al. analyzes the nature of proteins and their intrinsic properties (hydrophobicity, charge, presence in protein complexes) that are prone for aggregation upon heat shock. Their second question addresses the subsequent disaggregation reaction. Here, they studied the slope of disaggregation in the recovery period after heat shock and their correlation with intrinsic features. In addition, they analyzed the proteins that are synthesized in the recovery period in response to the heat shock. A similar analysis has been performed only in yeast cells, that utilize a different disaggregation chaperone complex (Hsp104). Thus, the data provided here on human cells are novel and interesting.

The manuscript is well written and the schematic illustration outlining the experiments help in understanding the set-up of the experiments.

Major comments:

1. This reviewer misses a discussion of the role of proteolysis in the clearance of protein aggregates by either autophagy or the UPS.
2. This study would greatly benefit from a parallel analysis upon depletion of at least one disaggregating chaperone such as Hsp110 (HSPH1, HSPH4), that is so far only described in vivo to facilitate the disaggregation reaction.
3. The authors noted that the disaggregation process is not completed at the 5 hour time point. Why did the authors then not choose an additional later time point in their analyses?
4. It may go beyond the scope of this study, but it would be interesting to compare the data with the aggregation propensity in response to other proteotoxic stresses that also allow a recovery such as oxidative stress.
5. The analysis of the synthesized chaperones in response to heat shock is very powerful for the identification of chaperones that could be part of the disaggregation complex. This should be further discussed. So far it is only known that 1 member each of the Hsp70, Hsp110 and J-domain protein family is required for the disaggregation of protein aggregates. Yet it is completely unknown which chaperone can functionally cooperate with which other chaperone(s). This study could actually help to shed some light on this question. E.g. one would expect that the disaggregating chaperones are co-regulated. In addition, the ideal stoichiometries among the chaperones have been established in in vitro assays. Can the authors use this information to speculate which chaperone complexes could be formed?

6. Did the authors identify an epitope within the aggregators (apart from the disordered region) that enables their interaction with chaperones? E.g. when analyzing all aggregating proteins - did the authors identify a motif?
7. This study would benefit if the authors could show a correlation (or delay) between disaggregation and the regain in function by assessing the functionality of one protein e.g. an enzyme before heat shock, upon heat shock and at the time points during recovery.
8. The authors put a huge emphasis of the correlation of aggregation and the disordered regions as well as random coil like structures (Fig 3 G+H), yet the correlation coefficient is with 0.25 and 0.19 very small. How meaningful are those correlations?

Minor comments:

1. The authors should discuss their choice of cell line. Why are K562 cells a suitable model? This is important as immortalized cancer cells are known to exhibit higher chaperone levels. It is obvious that such extensive proteome studies are not feasible in cultured cells that are not cancer cell lines. But at least the authors should justify their choice of cell type. What is known about them - how do they differ with regards to their proteome profile from HEK293 or HeLa cells that are conventionally used in cell culture experiments.
2. The authors should include 3 additional references that they missed. First, supersaturation of proteins was proposed to be a facilitator of protein aggregation (Ciryam et al., 2013 & 2015). Can the authors confirm this with their data set? Second, the observed decline in protein synthesis upon heat shock has been demonstrated before (Kirstein-Miles et al., 2013) and should be mentioned.
3. The figure presentation can be improved. E.g. the plots depicted for instance in figure 5 (B,C and D) are identical. The only difference is the labelling that highlights specific proteins and / or properties. It would be better to increase the figure size and show just one plot and by using different colors emphasize specific aspects. In addition, the labelling of the figures and heat maps could be improved in general by adding more description (also in the figure legends). Space can be saved by omitting the repeating illustrations of the experimental outline, which are similar anyway.
4. EV1 shows the viability of the cells in response to different lengths of heat exposure. Yet it is unclear how higher values than 1 can be obtained.
5. The manuscript could be more concise. There is a lot of repetition in the interpretation of the data e.g. "in other words" to explain inverse correlation. This is not necessary.
6. The buffer description of the lysis buffer is incomplete - the pH is missing.

Reviewer #2:

This manuscript from the Savitski group describes the effects produced by a stress insult (heat shock) on the proteome. Previous work in worm and mice models investigated if and how proteins aggregate or are differentially expressed under physiological conditions, multiple stresses and aging. Multiple processes were implicated in proteostasis maintenance with a common key involvement of the HSP70 chaperone system. In yeast the resolution of the aggregation of

endogenous proteins is instead more clearly associated with the action of the HSP104 and HSP100 disaggregases. While aggregation of proteins has been studied on a proteome-wide level with yeast and worm, especially in the context of aging and protein aggregation diseases, the authors attempt to fill a gap and zoom into the heat-induced aggregation and disaggregation of human proteins in a cell culture model (K562).

The manuscript can be essentially divided in two conceptual parts. In the first part the authors characterise the properties of a group of aggregating protein upon heat stress and their disaggregation dynamics after a recovery phase. They then measure the thermal stability of the proteins that remain soluble after heat shock. These two groups of proteins include DNA binding proteins and protein involved in the general maintenance of quality control. While this information is certainly valuable, the authors do not provide clear insights of what could this be the cellular strategy to invoke immediate protection against aggregation, based on the list of proteins that respond to heat shock.

Increasing our understanding of protein aggregation in human models is very relevant and would represent a step forward in understanding of proteostasis and aggregation related diseases. This work showcases multiple elegant and state-of-the-art proteomics experiments combining stability (thermal proteome profiling and TPP) and proteome dynamic (SILAC) approaches. While the application of multiplexed proteomics is very original and is definitely a major strength, the biological interpretation of the data and the potential novel biological insights are less remarkable, therefore the general aim of this work "To shed light on how human cells handle aggregates of endogenous proteins" does not fully deliver, at least in the current version of this manuscript. Perhaps the full potential of this dataset could become more noteworthy, if some experiments, definitions and technical details are better clarified.

MAJOR POINTS:

- The authors analyse different aspects of proteostasis after heat shock using TMT-labelling based proteomics. Are the experiments described in figure 1, figure 3, figure 4 and EV figure 2 independent, or do the different figures refer to different ways of analysing the data of the experimental schema shown in figure 1? Perhaps a different presentation of the experimental design could avoid this confusion.
- Was the protein aggregation / disaggregation analysis done quantifying only the pre-existing protein only (light channel)? If it is not the case, how did the authors calculate a correct FDR for peptide and then protein identifications, since two independent Mascot searches needed to be done to assign light and heavy peptides with the concomitant TMT labelling? Please clarify this point.
- In the section describing figure 2, the authors provide a working definition of "aggregators". Protein aggregators are the sub-population of proteins that present a differential solubility of approximately 0.66 between heat shocked and mocked shocked cells. Here, solubility is intended as the protein fraction that can be digested, thus soluble, in the lysis condition with mild detergent, which is supposedly non fully denaturant. When referring to figure EV2, another definition of solubility is provided as the fraction of protein measurable in NP40 buffer over SDS buffer. In figure EV2C proteins that show differential solubility between heat shocked and pre-shocked cells are also called "aggregators". Later in figure 3 the authors go back to the first definition of aggregation. This is confusing and a single, clearer definition of "aggregators" should be provided, since this is a key concept for many of the described observations.

- In addition to the previous point, I think that further care should be taken when discussing protein aggregates. Previous work from other groups (e.g. Hartl and Drummond laboratories, cited in this manuscript) adopt a different criteria to define aggregates, based on differential protein sedimentation after centrifugation. Others define aggregates when proteins form distinct foci in microscopy. Here differential solubility in NP40 and SDS is used, at least accordingly to one of the two principles used (see my previous point). The authors only very briefly comment about this existing literature noticing very limited overlap between with their data. Could they elaborate something more? Would their aggregating proteins be observable by microscopy? Perhaps, the lack of similarity might suggest that the definition used in this paper is limiting, or that different definitions of "aggregates" are not comparable.
- Along these lines... I am wondering whether a detergent based definition of protein aggregation could be at least partially the reason of the overrepresentation of DNA binding and nucleolus associated proteins in the "aggregators" category, especially considering that the aggregators largely coincide with the group of heat thermostable proteins found by TPP.
- The aggregators are also found to be enriched in hydrophilic and positively charged residues. This, together with the protein complexes membership, are properties that should also differentiate nuclear vs cytosolic proteins. Would you see the same enrichments, if you just did the same tests comparing nuclear and cytosolic proteins?
- Figure 3: Among the proteins that do not disaggregate after heat shock there are proteins involved in DNA damage. Is their lack of disaggregation compensated by a increase in new protein synthesis? Is this a general trend?
- The dynamic SILAC experiment done to analyse newly synthesized proteins reveals that the most upregulated proteins include many heat shock proteins. This is in line with what has been observed before in other whole proteome analysis of the heat shock response (including time courses). Perhaps the authors should highlight more what we learn from this new dynamic SILAC experiment in the context of heat shock.
- Dynamic SILAC experiment, figure 4C: here the increase in protein neo-synthesis was compared with published mRNA variations upon yeast stress. The authors here write that: "other proteins were upregulated only at the protein level (Figure 4G), suggesting that their upregulation is translation rather than transcription driven". Considering that the majority of upregulated proteins of figure 4G are also upregulated at the mRNA level (7/11), I would mitigate this statement.
- I find interesting that the soluble fraction of many "aggregators" proteins become more thermally stable. Perhaps this suggests that there are subpopulations of proteins with different melting temperature. What would be the outcome of this experiment if the heavy labelled (newly synthesized proteins) are analysed ?
- In the thermal stability experiment I did not quite understand why the heat shock-induced aggregation is adjusted with changes measured at 37 and 37.8 degrees, if the heat shock was done at 44 degrees. The reference to the material and methods does not address this.

MINOR POINTS

- The order of the panels of the figures is not always consistent with the text. Please correct.

- The authors mentioned that they performed initial control experiments to ensure that the intracellular pool of arginines and lysines was consumed from the light SILAC medium before heat shock. These experiments could be reported as supplementary information.
- Figure EV2B: If the purpose of this figure is to show that the total protein amount remains constant over 10 minutes of heat shock, why are the correlation plots relative to the recovery time points also shown?
- The GO enrichment analysis for the aggregators should specify the statistics applied and the statistical significance cut off applied.

Reviewer #3:

The manuscript by Määttä and coworkers analyzed the aggregation and disaggregation of human proteins under heat stress using mass spectrometry-based proteomics in combination with dynamic SILAC and TMT multiplexing. They first defined aggregating proteins by their absence in the soluble fraction upon mild heat stress compared to control. They found the proteins in the analyzed aggregates to have on average distinct physicochemical properties (higher disorder, higher molecular mass, higher iP and more hydrophilic AA content) compared to non-aggregating proteins and that proteins in the aggregates are more likely part of large multi-protein complexes. Second, they report that aggregated proteins disaggregated over time, with proteins with higher disorder showing faster aggregation and disaggregation rates. Finally, the authors provide 2D-TPP data to show that a sub-population of aggregating proteins remains soluble after heat shock and shows increased thermal stability afterwards.

Overall, this is an interesting dataset based on state-of-the-art mass spec technology. While the presented data analysis and the concepts discussed are interesting, there are a number of points the authors need to address before the paper could become suitable for publication. First, there is no independent experimental validation of the mass spec data. I would have expected to see at least some immunofluorescence data on the subcellular distribution of proteins before and after heat shock. This would also indicate which type of "aggregates" actually mediates the solubility changes they observed. Second, data analysis and discussion misses important aspects related to disordered proteins and their role in protein-protein interactions and protein toxicity. Third, the authors fail to analyze/discuss the possibility that the heat-shock induced thermal stabilization of proteins could just reflect different cellular pools of the same protein. Also, the data is currently inaccessible for reviewers and no supplemental tables are provided. Therefore, while I expect the data to be of high quality given the descriptions provided and the excellent track record of the lab, there is no way for me to assess this.

Specific points:

1. There is no independent experimental validation of the data. I think immunofluorescence microscopy with antibodies against a number of aggregating proteins could be used to (i) validate the MS data and (ii) show how and where the proteins aggregate and disaggregate.
2. There is considerable data on the role of intrinsic protein disorder in dosage sensitivity (Vavouri et al., Cell, 2009; DOI: 10.1016/j.cell.2009.04.029), liquid-liquid phase separation (Bolognesi et al., Cell Rep., 2016, DOI: 10.1016/j.celrep.2016.05.076), protein complex formation (McShane et al., Cell, 2016, DOI: 10.1016/j.cell.2016.09.015) and aggregation during aneuploidy (Brennan et al., Genes Dev,

2019, doi: 10.1101/gad.327494.119). My main critique is that the findings should be analyzed and discussed in the light of these previous findings. For example, since disorder appears to be a key determinant of dosage sensitivity (Vavouri et al.) it would make sense to sequester them in aggregates upon overexpression, which indeed has been shown (Brennan et al.). Also, given the potentially toxic nature of disordered proteins, it would make sense to make sure that they are not made in excess relative to their interaction partners, which was also shown (McShane et al.). Heat shock might liberate disordered proteins from existing complexes or disrupt de novo assembly of complexes from newly synthesized proteins, which in both cases could lead to cellular overabundance of disordered proteins. In this context, it could also be interesting to look at differences in the aggregation propensity of preexisting and newly synthesized proteins, which should be possible in the dynamic SILAC dataset.

3. Related to the observed changes in thermal stability after heat shock, I am wondering how much of this simply reflects different protein pools. For example, assuming that the free fraction of DNA-binding proteins is more sensitive to heat-induced aggregation than the DNA-bound fraction, we would expect that applying the heat shock depletes the free fraction and selects the DNA-bound fraction, which will then appear stabilised in the subsequent TPP experiment. However, the thermal stability of the proteins is actually not changing. Rather, TPP on mock-shocked and heat-shocked cells involves looking at different subpopulations of protein molecules which already had different thermal stability before the heat shock. This also relates to the discussion section: The authors write "it is tempting to speculate that chromatin binding could also have protein quality aspects by stabilizing and sequestering unstable DNA-binding proteins during proteotoxic stress". The more parsimonious explanation is that the DNA-bound fraction is (i) generally more thermally stable and (ii) less likely to aggregate upon heat shock (note that these two points are probably also biophysically related to each other). Applying the heat shock merely selects for the DNA-bound fraction of a protein, which is why it looks more thermally stable. There is no need to speculate about a possible role of chromatin in protein quality control. More generally, proteins can exist in different "states" in the cell (different interaction partners, proteoforms, PTMs) that have different thermostability and correlated differences in aggregation propensity.

4. The proteomic data has to be made available to reviewers (both the raw files and processed files).

5. While the light labeled proteins are in steady state condition, the heavy channel contains newly synthesized proteins (after 90 mins). This makes the interpretation of the difference in solubilities between subunits within a complex more complicated. The authors cannot distinguish between free non-assembled subunits and assembled subunits. At 90 mins the assembly of many complexes will not be finished. Thus, the signal is a mixture of both protein pools (unassembled and assembled). Proteins incorporated into a complex likely have different melting curves compared to their free counterparts (Tan et al. (2018), Science).

6. Aneuploidy has recently been shown to increase protein aggregation in yeast (Brennan et al. 2019). Since K562 are aneuploid the authors may want to reanalyze their data in this context (enrichment for proteins encoded in genomically amplified regions). Aneuploidy might help explain some of the observed aggregation before heat stress.

7. "Directly after heat shock, the abundance of 300 proteins (<7% of quantifiable proteome) decreased significantly (Benjamini-Hochberg adjusted p-value < 0.05 in LIMMA analysis and fold change < 2/3) in the soluble fraction when compared to mock shocked sample (Fig 2A)." It is not clear to me off this statement refers to the preexisting (light) proteins or the newly synthesised (heavy) ones. This should be clarified.

8. Figure 2I: It took me a while to understand this figure because of the missing column labels. The rows are clearly labeled with the subunits but the columns are not. I assume the columns also represent complex subunits. This should be indicated for clarification.

9. "Together, these results indicate that the disaggregation is the preferred method to deal with

aggregates." The word "preferred" implies that disaggregation is somehow advantageous, which cannot be concluded from the data in figure 3 B and C.

10. "we found a significant (Benjamini-Hochberg adjusted p-value < 0.05) negative correlation between disaggregation slope and the loss of solubility after heat shock (Fig 3F; $p = 1.23 \times 10^{-13}$; $r = -0.42$). In other words, the more a protein aggregated, the faster it disaggregated during the recovery." I am wondering if there might be a technical reason for this observation. In essence, the observation is that some proteins show larger changes (during aggregation and disaggregation) and others smaller ones. Could this be related to systematically different degrees of TMT ratio compression?

11. Certain subgroups of proteins (transcription factors, DNA-binding proteins, perhaps ribosomal proteins) are overrepresented in the aggregated proteins. This begs the question whether molecular features associated with aggregation/disaggregation (disorder, amino acid composition) are truly features of aggregation/disaggregation or merely features of these protein subgroups. I realise this is a chicken and egg question that is hard to answer. But it might be possible to take a look at two distinct subgroups of aggregators (such as DNA-binders versus non-DNA binders) in order to see if they share similar molecular features or not.

12. "Within a protein complex ($n = 32$), aggregators had more similar disaggregation profiles when compared to scrambled complexes ($n = 10000$) containing the same aggregators randomly redistributed (Fig EV4A; $p = 0.048$)." It should be stressed that this signal is quite weak. Also, I assume that taking more than 10000 scrambled complexes would make the p-value more significant. Hence, the functional significance of the nominally significant p-value is not clear to me.

13. "Interestingly, from our ten most upregulated proteins, only heat shock proteins were also upregulated on mRNA level; other proteins were upregulated only at the protein level (Fig 4G) suggesting that their upregulation is translation rather than transcription driven." Alternatively, the observed protein-level up-regulation could reflect decreased protein degradation. No direct evidence for translational control is presented.

14. "Therefore, we speculate whether disordered regions could serve other functions not related to sequestering proteins to aggregates. Disordered regions could offer, for example, shielding for potentially toxic protein-protein interactions, such as seeding and formation of amyloid fibers." As mentioned in point 1, disorder itself appears to be toxic, and cells generally tend *not* to overproduce disordered proteins that are part of multiprotein complexes. Therefore, the idea that disordered regions could somehow "shield" toxic protein-protein interactions does not sound plausible to me.

We thank the reviewers for their insightful comments, that we feel have greatly improved the manuscript.

Reviewer #1:

Summary:

The manuscript by Määttä et al. analyzes the nature of proteins and their intrinsic properties (hydrophobicity, charge, presence in protein complexes) that are prone for aggregation upon heat shock. Their second question addresses the subsequent disaggregation reaction. Here, they studied the slope of disaggregation in the recovery period after heat shock and their correlation with intrinsic features. In addition, they analyzed the proteins that are synthesized in the recovery period in response to the heat shock. A similar analysis has been performed only in yeast cells, that utilize a different disaggregation chaperone complex (Hsp104). Thus, the data provided here on human cells are novel and interesting.

The manuscript is well written and the schematic illustration outlining the experiments help in understanding the set-up of the experiments.

Major comments:

1. This reviewer misses a discussion of the role of proteolysis in the clearance of protein aggregates by either autophagy or the UPS.

We strongly agree with the importance of protein degradation in aggregate clearance. We realize that the focus on the manuscript has been much on disaggregation and agree that the degradation aspect should also be discussed.

Action taken:

- *We added a paragraph in the introduction where we briefly discuss the different ways cells handle protein aggregates (including protein degradation):*
 - *“Cells have multiple ways to handle protein aggregates [16-18]. Irreversibly damaged proteins can be degraded by the ubiquitin-proteasome system [19]. Larger aggregates can be cleared by autophagy or secreted out from the cells [18]. To maintain protein homeostasis, protein degradation can be balanced by upregulated protein synthesis. In addition, protein disaggregation and re-folding allows to rescue functional proteins from the aggregates [16, 20, 21].” [Page 2. Paragraph 3]*

- *We added discussion about the absence of degradation that we (and Wallace et al. 2015, DOI: 10.1016/j.cell.2015.08.041) observed during recovery:*
 - *“We observed that the majority of aggregated proteins disaggregated during recovery (Fig 3A-B). These findings indicate that, as observed with yeast [12], the main strategy for human cells to handle aggregates is disaggregation. However, it should be noted that proteasomal degradation, as well requires aggregate re-solubilization prior to degradation [21]. Therefore, it remains an open question whether the disaggregated proteins are destined for re-folding or degradation. Based on our data, at least in the five hours following heat shock no degradation was observed (Fig EV4A).” [Page 13. Paragraph 4]*

2. This study would greatly benefit from a parallel analysis upon depletion of at least one disaggregating chaperone such as Hsp110 (HSPH1, HSPH4), that is so far only described in vivo to facilitate the disaggregation reaction.

We appreciate noticing the potential of our method to answer additional questions. Our main goal was to study proteome-wide disaggregation in human cells since this type of experiment has not been reported before. While we agree that it would be very exciting to see how protein disaggregation patterns would change upon Hsp110 depletion, we feel that this is outside the initial scope of the manuscript.

After reviewing the comments from the reviewers and editor, we decided to put our experimental efforts on validating the mass spectrometry results with imaging rather than expanding the method we introduced.

3. The authors noted that the disaggregation process is not completed at the 5 hour time point. Why did the authors then not choose an additional later time point in their analyses?

Our experimental design was based on the assumption that disaggregation would be completed in less than an hour. This assumption was based on results from in vitro (e.g. Nillegoda et al. 2015, DOI: 10.1038/nature14884) studies and the only reported proteome-wide disaggregation study so far (Wallace et al. 2015, DOI: 10.1016/j.cell.2015.08.041).

We had to update our design a few times. At first, we tried time intervals of few minutes for the recovery but did not observe clear signs of disaggregation. Then, we tried tens of minutes and could observe a disaggregation trend, and we finalized the design by having the time points reported in the manuscript. It was a surprise for us to observe that even after five hours, the disaggregation was not complete. We made the decision to continue with the data we had collected rather than repeating the experiments, considering the costs involved in performing these types of experiments as we could already detect and analyze the disaggregation from the existing data.

4. It may go beyond the scope of this study, but it would be interesting to compare the data with the aggregation propensity in response to other proteotoxic stresses that also allow a recovery such as oxidative stress.

We think this would be an excellent idea. Developing proteomics experiments to compare aggregation and disaggregation characteristics in different stress conditions would be very interesting extension to our study. Reports with non-human model organisms that take this kind of approach have made some interesting findings related to protein aggregation in different stress conditions; for example, Weids et al. (2016, DOI: 10.1038/srep24554) and Sui et al. (2020, DOI: 10.1073/pnas.1912897117).

Action taken:

- *We added a discussion about adapting our methods to study other stress conditions:*
 - *“While we introduced these techniques here to study heat-induced solubility changes, in the future it would be interesting to adapt the same approach to study and compare different stress condition that lead to protein aggregation such as chemical or other environmental stress conditions.” [Page 14. Paragraph 4; Page 15, Paragraph 1]*

5. The analysis of the synthesized chaperones in response to heat shock is very powerful for the identification of chaperones that could be part of the disaggregation complex. This should be further discussed. So far it is only known that 1 member each of the Hsp70, Hsp110 and J-domain protein family is required for the disaggregation of protein aggregates. Yet it is completely unknown which chaperone can functionally cooperate with which other chaperone(s). This study could actually help to shed some light on this question. E.g. one would expect that the disaggregating chaperones are co-regulated. In addition, the ideal stoichiometries among the chaperones have been established in in vitro assays. Can the authors use this information to speculate which chaperone complexes could be formed?

This is a very interesting suggestion. Indeed, we have noticed that from the list of most up-regulated proteins one could theoretically build up a Hsp70 chaperone system/disaggregase. The most strongly upregulated proteins include a Hsp70 (HSPA1A| HSPA1B), nucleotide exchange factor (HSPH1) and two Hsp40s (DNAJB1 and DNAJB4). We could extend our speculation based on Nillegoda et al. (2015, DOI: 10.1038/nature14884) that the highly upregulated Hsp40s from class B are involved in disaggregation of larger aggregates - as opposite to class A Hsp40s that favor smaller aggregates. Although, other Hsp40s are upregulated as well. These include, for example, DNAJC8 that has no obvious link to disaggregation.

Action taken:

- We added a new figure (Appendix Figure 9S) that shows the synthesis rates for different Hsp40s after heat shock:

- We added a description of the upregulation of class B Hsp40s and their role in disaggregation:
 - “Interestingly, both highly upregulated Hsp40s (DNAJB1 and DNAJB4) belonged to class B of Hsp40s which are involved in protein disaggregation [27]. Although other Hsp40s were also upregulated (Appendix Figure 9S), this suggested that the response to heat shock involved an increase in disaggregation capacity by upregulated protein synthesis.” [Page 9. Paragraph 6; Page 10. Paragraph 1]
- We added a discussion related to the possible co-regulation of disaggregase components:
 - “We noted that the most upregulated heat shock proteins could, in theory, form a Hsp70-based disaggregase if assuming that the components would be co-expressed. These proteins include a Hsp70 (HSPA1A| HSPA1B) with a nucleotide exchange factor (HSPH1) and two Hsp40 proteins (DNAJB1 and DNAJB4). To speculate further, the two class B Hsp40s could reflect an adjustment towards clearance of larger aggregates, as it has been suggested for class B in contrast to class A (which are involved in clearing smaller size aggregates) [27]. However, it should be noted that other Hsp40s are upregulated too, for example DNAJC8, that has no known link to disaggregation. Further studies would help to understand the co-expression of disaggregase components and the possible adjustment of its composition to reflect the aggregation load.” [Page 14. Paragraph 2]

6. Did the authors identify an epitope within the aggregators (apart from the disordered region) that enables their interaction with chaperones? E.g. when analyzing all aggregating proteins - did the authors identify a motif?

We had not previously searched for common motifs within aggregators and are very grateful for the idea.

As far as we know, most chaperones do not have a binding motif. However, it has been recognized that Hsp70 has a preference for stretches of four to five hydrophobic amino acids that are flanked by positively charged residues (Rudiger, et al. 1997, DOI: 10.1093/emboj/16.7.1501). We looked for these binding motifs (based on Rudiger et al.) and found them in both aggregators and soluble proteins. We observed a slightly higher, yet not significantly enriched, occurrence of such binding motifs in aggregators versus soluble proteins (panel A in the figure below). In addition, we analyzed the amino acid content within the motifs and could not find major differences between them (panel B in the figure below); the high leucine content in the motif with five hydrophobic amino acids in aggregators is based on only five proteins and we would hesitate to make strong conclusions from it.

Action taken:

- We added a new figure (Appendix Figure 4S) where we show the results of our motif analysis:

- We added a description of the results:
 - “To analyze whether aggregators would have stronger preference for chaperones we searched for Hsp70 binding motifs in them. A Hsp70 binding motif has been reported to contain four or five hydrophobic residues flanked by positively charged residues [52]. We found a slightly higher occurrence of the binding motifs in aggregators but the difference

was not significant ($p = 0.09582$, Appendix Figure 4SA). In addition, we could not find differences in the amino acid composition within the binding motif of aggregators as compared to the soluble proteins (Appendix Figure 4SB).” [Page 6. Paragraph 4]

7. This study would benefit if the authors could show a correlation (or delay) between disaggregation and the regain in function by assessing the functionality of one protein e.g. an enzyme before heat shock, upon heat shock and at the time points during recovery.

This would be an interesting experiment. Considering that Wallace et al. (2015, DOI: 10.1016/j.cell.2015.08.041) found that aminoacyl transferase is active in the aggregates it would be interesting to follow the activity of enzymes upon heat shock and recovery.

As stated in our reply to a previous comment, considering all the responses we got from the reviewers, we decided to put our experimental efforts in validating the mass spectrometry results with microscopy.

8. The authors put a huge emphasis of the correlation of aggregation and the disordered regions as well as random coil like structures (Fig 3 G+H), yet the correlation coefficient is with 0.25 and 0.19 very small. How meaningful are those correlations?

We agree that the correlation coefficients are relatively small. However, we extended the correlation analysis by applying statistical testing. In the test, we analyze how infrequent the observed correlations would be in a null hypothesis scenario where no correlation exists.

The results from this statistical test indicate that the probability of finding these correlations by chance would be less than 0.008% for disordered regions and less than 0.5% for random coil (p -values adjusted for multiple hypothesis testing).

In addition, we think that it is highly unlikely that the two correlations that are two orthogonal measures of the same protein property would stand out in the correlation analysis merely by chance.

Action taken:

- *We have modified the text to reduce the emphasis and discuss the low correlation coefficients:*

- *In the abstract, we have changed*

*“ . . . disordered regions **also resulted in** faster disaggregation . . . ”*

to

*“ . . . disordered regions **were associated with** faster disaggregation . . . ”*

[Page 1. Abstract]

- *When summarizing disaggregation results, we have changed*

*“In addition, larger extent of intrinsically disordered regions in a protein **resulted in** faster disaggregation.”*

to

*“In addition, larger extent of intrinsically disordered regions in a protein **associated with** faster disaggregation.” [Page 9. Paragraph 1]*

- *From the abstract, we have removed the statement*

“. . . [Our results] propose novel roles for intrinsically disordered regions in protein quality control. . .”

[Page 1. Abstract]

- *We have added a statement that the correlation is “weak but significant” when we describe the results [Page 8. Paragraph 3]*

Minor comments:

1. The authors should discuss their choice of cell line. Why are K562 cells a suitable model? This is important as immortalized cancer cells are known to exhibit higher chaperone levels. It is obvious that such extensive proteome studies are not feasible in cultured cells that are not cancer cell lines. But at least the authors should justify their choice of cell type. What is known about them - how do they differ with regards to their proteome profile from HEK293 or HeLa cells that are conventionally used in cell culture experiments.

We think that the high chaperone levels in cancer cells is an important aspect to consider and believe it has some implications in aggregation as well as disaggregation. It would be interesting to speculate on how much of the higher chaperone levels are used by the cancer cells and would there be an increase in “free” chaperone capacity.

We had two main reasons to choose K562 cells. First, K562 is a thermotolerant cell line. They can be treated, for example, ten minutes at 45C without any loss of viability [e.g., Mivechi 1989 (PMID: 3378207) and our data in Figure EV1]. Since we aimed to study recovery from heat shock it was very important for us to have cells that would survive the heat treatment. Second, we needed a cell line that is easy to handle. This becomes an important issue since our experimental design includes steps where cells are transferred. At the beginning – although we do not mention this in the manuscript – we started the experiments with recovery times of few minutes which demanded the transfer of cells to be very fast. Therefore, we needed cells that grow in suspension.

Action taken:

- We added a statement about why we chose K562 cells to Materials and Methods:

“The K562 cells were chosen based on their tolerance to a 45°C heat shock [Mivechi 1989] (Figure EV1) and easy handling (suspension cells).” [Page 17. Paragraph 1]

2. The authors should include 3 additional references that they missed. First, supersaturation of proteins was proposed to be a facilitator of protein aggregation (Ciryam et al., 2013 & 2015). Can the authors confirm this with their data set? Second, the observed decline in protein synthesis upon heat shock has been demonstrated before (Kirstein-Miles et al., 2013) and should be mentioned.

We analyzed our data with the algorithm presented in Ciryam et al. 2013 and could not find a difference in the supersaturation score between aggregators and soluble proteins (panel A in the figure below). However, the structurally corrected Zyggregator algorithm predicted correctly that the aggregators were more prone for aggregation (panel B in the figure below).

We agree with the reviewer on the second point.

Action taken:

- We added a new figure (Appendix Figure 5S) where we show the result from analyzing supersaturation of aggregators:

- We added a description of the results from the analysis of supersaturation:
 - *“Supersaturation (high concentration relative to solubility) has been shown to correlate with aggregation propensity [Ciryam et al. 2013, Ciryam et al. 2015]. We*

found that supersaturation scores [Ciryam et al. 2013] were not higher for aggregators than for soluble proteins (Appendix Figure 5SA). The supersaturation score has two components: protein concentration and structurally corrected aggregation propensity score (based on Zyggregator method [Tartaglia et al. 2008]) [Ciryam et al. 2013]. While the supersaturation score was not higher for aggregators, we found a higher Zyggregator-based aggregation propensity score for them (Appendix Figure 5SB). This suggests that the aggregation propensity in heat shock is more related to structural features of proteins rather than to supersaturation.” [Page 6. Paragraph 5]

- *We have added the suggested reference (Kirstein-Miles et al. 2013) as well as others to the manuscript and included a statement that the decline in protein synthesis is in line with previous reports. [Page 9. Paragraph 4]*

3. The figure presentation can be improved. E.g. the plots depicted for instance in figure 5 (B,C and D) are identical. The only difference is the labelling that highlights specific proteins and / or properties. It would be better to increase the figure size and show just one plot and by using different colors emphasize specific aspects. In addition, the labelling of the figures and heat maps could be improved in general by adding more description (also in the figure legends). Space can be saved by omitting the repeating illustrations of the experimental outline, which are similar anyway.

We agree with the reviewer about the identical presentation in Figure 5C-D and have now compressed the results to one panel.

Action taken:

- *Panels B-D in Figure 5 are merged to a one panel B.*
- *The style of the new panel B was modified for visual clarity*
 - *Coloring of stabilized and destabilized proteins swapped*
 - *Only the strongly stabilized MAPK kinases of the stress-activated MAPK pathway are annotated*
 - *Aggregators are highlighted as smaller and darker points*
 - *The number of aggregators was removed*
 - *The color and shape of data points for proteins from stress-activated MAPK pathway is changed*

Since the reviewer made a general suggestion about adding more description to figures and figure legends, we added them to places we thought could benefit from them based on the comments from all reviewers. We hope that the additions have improved the clarity of the manuscript.

Action taken:

- *Annotation for SILAC fraction added to all relevant panels in figures 2, 3, 4 (also added to figure legend), EV2, EV3 and EV4.*
- *As suggested also by another reviewer, more protein annotations added to a heat map. [Figure 2I]*
- *Colors changed to the experimental design in Figure EV2A to make the distinction to the design in Figure 1A more clear.*
- *The repetitive experimental designs in Figure 2A, 3A and 4A are removed.*
- *Related to previous point, new panel EV2B was introduced that includes TMT labelling scheme for the experiment described in Figure EV2A.*
- *Thorough panel re-arrangements have been made to Extended View figures 2-4 to have the figures follow the text flow:*
 - *Panel B in the original Figure EV2 that shows protein intensities in SDS-lysed samples from all time points and both SILAC fractions has been separated to three panels*
 - *Values after heat shock (light SILAC) is now Figure EV2C*
 - *Values during recovery (light SILAC) is now Figure EV4A*
 - *Values after heat shock and during recovery (heavy SILAC) is now Figure EV4D*
 - *Protein complex analysis of disaggregation in Figure EV4A is moved to figure EV4C*
 - *Protein complex analysis of aggregation in Figure EV4B is moved to Figure EV2G*
 - *Figure EV2E showing comparing disaggregation slopes and the presence of insoluble protein fraction is moved to Figure EV4B*

We agree that the repeated illustrations of the experimental outline could be removed without losing clarity.

Action taken:

- *Experimental outline and equation for solubility calculation removed from panel A. [Figure 2]*
- *Panel A showing experimental outline and equation for solubility calculations removed [Figure 3]*
- *Panel A showing experimental outline removed [Figure 4]*
- *Remaining panels re-positioned to balance the changes. [Figure 2, Figure 3, Figure 4]*
- *Panel A description in figure legend removed. Panel names updated in figures, figure legends and text. [Figure 3, Figure 4]*

4. EV1 shows the viability of the cells in response to different lengths of heat exposure. Yet it is unclear how higher values than 1 can be obtained.

The higher than 1 values for the viability comes from variability in the data. This seems exaggerated as the data has been represented as a relative measure of viability compared to the 37°C sample. In order to improve clarity and show that the observed values are within the measurement errors of the assay, we

now show the raw data of the viability. It is noteworthy that such a representation also captures the trend of decreasing cellular viability at higher temperatures more clearly.

Action taken:

- Figure EV1 updated to show raw data instead of 37°C-normalized values.
- A statement “Intensities relative to a control treatment at 37°C are shown for each replicate.” removed from the legend of Figure EV1.
- A statement “. . . and values were normalized to sample with heat treatment at 37C” removed from materials and methods [Page 22. Paragraph 2]

5. The manuscript could be more concise. There is a lot of repetition in the interpretation of the data e.g. "in other words" to explain inverse correlation. This is not necessary.

We agree that the interpretations are repeated in certain places in the manuscript. Hence, we have made the following changes to make the manuscript clearer and concise.

Action taken:

- We have removed the repeating parts from the following statements (deletions in strikethrough):
 - ~~“In other words, aggregators were enriched in hydrophilic residues as well as diminished from hydrophobic residues (Fig 2D).”~~ [Page 5. Paragraph 2]
 - ~~“In other words, the more a protein aggregated, the faster it disaggregated during the recovery.”~~ [Page 8. Paragraph 3]
 - ~~“Therefore, a higher amount of disordered regions in proteins related to faster disaggregation.”~~ [Page 8. Paragraph 3]
- A sentence that included repetition was removed and rewritten to improve clarity as requested by another reviewer (deletions in strikethrough and additions in bold):
 - **“The extent of loss of solubility after heat shock was comparable between the two types of aggregators (Fig EV2E).** ~~The two types of aggregators had similar solubility changes after heat shock, i.e., the solubility at physiological conditions did not determine the extent of aggregation upon heat shock (Fig EV2D).~~” [Page 4. Paragraph 5]

6. The buffer description of the lysis buffer is incomplete - the pH is missing.

We thank the reviewer for pointing out the missing information. The pH is now added to the lysis buffer description.

Action taken:

- *pH added to the lysis buffer description [Page 19. Line 756; Page 21. Line 806]*

Reviewer #2:

This manuscript from the Savitski group describes the effects produced by a stress insult (heat shock) on the proteome. Previous work in worm and mice models investigated if and how proteins aggregate or are differentially expressed under physiological conditions, multiple stresses and aging. Multiple processes were implicated in proteostasis maintenance with a common key involvement of the HSP70 chaperone system. In yeast the resolution of the aggregation of endogenous proteins is instead more clearly associated with the action of the HSP104 and HSP100 disaggregases. While aggregation of proteins has been studied on a proteome-wide level with yeast and worm, especially in the context of aging and protein aggregation diseases, the authors attempt to fill a gap and zoom into the heat-induced aggregation and disaggregation of human proteins in a cell culture model (K562).

The manuscript can be essentially divided in two conceptual parts. In the first part the authors characterise the properties of a group of aggregating protein upon heat stress and their disaggregation dynamics after a recovery phase. They then measure the thermal stability of the proteins that remain soluble after heat shock. These two groups of proteins include DNA binding proteins and protein involved in the general maintenance of quality control. While this information is certainly valuable, the authors do not provide clear insights of what could this be the cellular strategy to invoke immediate protection against aggregation, based on the list of proteins that respond to heat shock.

Increasing our understanding of protein aggregation in human models is very relevant and would represent a step forward in understanding of proteostasis and aggregation related diseases. This work showcases multiple elegant and state-of-the-art proteomics experiments combining stability (thermal proteome profiling and TPP) and proteome dynamic (SILAC) approaches. While the application of multiplexed proteomics is very original and is definitely a major strength, the biological interpretation of the data and the potential novel biological insights are less remarkable, therefore the general aim of this work "To shed light on how human cells handle aggregates of endogenous proteins" does not fully deliver, at least in the current version of this manuscript. Perhaps the full potential of this dataset could become more noteworthy, if some experiments, definitions and technical details are better clarified.

MAJOR POINTS:

- The authors analyse different aspects of proteostasis after heat shock using TMT-labelling based

proteomics. Are the experiments described in figure 1, figure 3, figure 4 and EV figure 2 independent, or do the different figures refer to different ways of analysing the data of the experimental schema shown in figure1? Perhaps a different presentation of the experimental design could avoid this confusion.

We apologize for the confusion. All the results in the Figures 2-4 were based on the same experiment introduced in Figure 1. The experiment introduced in Figure EV2 is an independent experiment. We have made the following modifications to improve the clarity of these issues.

Action taken:

- *As requested by another reviewer, we have removed the repeating experimental outlines in panel A of Figures 2-4.*
 - *We changed the coloring of experimental design in Figure EV2A to make it distinct from the design shown in Figure 1A.*
 - *We introduced a new panel to Figure EV2 (panel B) showing the TMT labelling scheme of the experiment. The description of the new panel was added to the figure legend.*
-
- Was the protein aggregation / disaggregation analysis done quantifying only the pre-existing protein only (light channel)? If it is not the case, how did the authors calculate a correct FDR for peptide and then protein identifications, since two independent Mascot searches needed to be done to assign light and heavy peptides with the concomitant TMT labelling? Please clarify this point.

Yes. The aggregation and disaggregation analysis was performed only for the pre-existing proteins (light). This information was given in

- *the figure legend of Figure 2 (aggregation):*
“Data shown for pre-existing protein fraction (light) quantified with at least two unique peptides from at least two biological replicates.”
- *the figure legend of Figure 3 (disaggregation):*
“Data shown in B-H for pre -existing protein fraction (light) quantified with at least two unique peptides from at least two biological replicates.”
- *the main text before describing disaggregation results:*
“To monitor protein solubility during recovery, we measured protein intensities of the pre-existing proteins (light) in the soluble (NP40 extractable) fraction.”

This and other comments highlight the fact that we should have made the SILAC fraction under analysis more apparent in the manuscript. Hence, we now explicitly state it in the figures.

Action taken:

- *We added text labels to all relevant panels in Figures 2-4 and EV2-4 to indicate the SILAC fraction under analysis*
- *Before starting to describe the aggregation results in the main text, we added a statement*
“The protein aggregation was analyzed from the pre-existing (light) fraction” [Page 3. Paragraph 6]

• In the section describing figure 2, the authors provide a working definition of "aggregators". Protein aggregators are the sub-population of proteins that present a differential solubility of approximately 0.66 between heat shocked and mocked shocked cells. Here, solubility is intended as the protein fraction that can be digested, thus soluble, in the lysis condition with mild detergent, which is supposedly non fully denaturant. When referring to figure EV2, another definition of solubility is provided as the fraction of protein measurable in NP40 buffer over SDS buffer. In figure EV2C proteins that show differential solubility between heat shocked and pre-shocked cells are also called "aggregators". Later in figure 3 the authors go back to the first definition of aggregation. This is confusing and a single, clearer definition of "aggregators" should be provided, since this is a key concept for many of the described observations.

We completely agree with the importance of having only one clear definition for aggregators. The one definition for aggregators given in Figure 2A was indeed being used throughout the manuscript and in every figure. We named the “new definition” in Figure EV2C (based on NP-40 and SDS intensities) as “insoluble sub-population at physiological conditions” with the aim to avoid confusion. Thank you again for pointing this out. We should have been more clear about the issue.

Action taken:

- *We updated the legend of Figure EV2 as follows (updates in bold):*

*“. . . Aggregators (**as defined in Fig 2A**) are highlighted with darker color. **In addition,** proteins were assigned to contain an insoluble sub-population at physiological conditions if the total solubility of the pre-shocked sample was lower than -0.6 (**highlighted with cyan at the left of the figure**).”*

- *We updated the main text where we discuss the results related to insoluble sup-populations (additions in bold and deletions in strikethrough):*

- *“Comparison of the ratio of NP-40-extracted (soluble sub-population) and SDS-extracted (total) proteins reports on the solubility status of a protein. Under physiological conditions, proteins involved in phase separated membrane-less nuclear organelles—such as the nucleolus—have been shown to contain an insoluble sub-population [44, 45]. We found that aggregators included many such proteins in addition to aggregators that were completely soluble in unstressed conditions Analysis of NP-40/SDS ratio of pre-existing protein pool between pre-shock and heat shock conditions showed that the aggregators included proteins that were completely soluble as well as proteins that have an insoluble sup-population under physiological conditions (Fig EV2D). The two types of aggregators had similar solubility changes after heat shock The extent of loss of solubility after heat shock was comparable between the two types of aggregators (Fig EV2E). We also observed that proteins from the cytosolic ribosome had an insoluble fraction in unstressed conditions (Fig EV2F). Similar observations have been made in yeast [13]. We speculate that this insoluble fraction represents ribosomal proteins in the nucleolus, where the ribosomes are assembled. Contrary to cytosolic ribosomes, we found that proteins from mitochondrial ribosomes are fully soluble in unstressed conditions (Fig EV2F).” [Page 4. Paragraph 5]*

- In addition to the previous point, I think that further care should be taken when discussing protein aggregates. Previous work from other groups (e.g. Hartl and Drummond laboratories, cited in this manuscript) adopt a different criteria to define aggregates, based on differential protein sedimentation after centrifugation. Others define aggregates when proteins form distinct foci in microscopy. Here differential solubility in NP40 and SDS is used, at least accordingly to one of the two principles used (see my previous point). The authors only very briefly comment about this existing literature noticing very limited overlap between with their data. Could they elaborate something more? Would their aggregating proteins be observable by microscopy? Perhaps, the lack of similarity might suggest that the definition used in this paper is limiting, or that different definitions of "aggregates" are not comparable.

We strongly agree that there are many reported criteria for aggregation; it is a very important topic since it has caused a lot of confusion as discussed, for example, by Mogk et al. (2018, DOI: 10.1016/j.molcel.2018.01.004). As being completely aware of this issue, we had a clear definition of what we mean by aggregation in the very beginning of the results section.

We have added - as requested by another reviewer - a microscopy experiment where we followed protein localization upon heat shock (and during recovery). It was found that aggregators (HELLS, BRD4 and TARDBP) had a small fraction that translocated from nucleus (where they localized in control samples) to cytoplasm and HELLS formed foci upon heat shock. In addition, the analyzed set of aggregators include disaggregating (BRD4 and TARDBP) as well as non-disaggregating (HELLS) proteins. After five hours of recovery, the cytoplasmic intensity had returned to control levels for the disaggregating proteins while the non-disaggregating HELLS still had cytoplasmic foci after the recovery. The localization of HDAC1, a nuclear protein that did not aggregate in the mass spectrometry experiment, had no observable change in the microscopy experiment upon heat shock. Therefore, the aggregation we defined in the beginning of the results section coincide with transport of the protein to cytoplasm and foci formation.

Action taken:

- *Results from immunofluorescence microscopy are presented in Figure 3D and Appendix Figure 6S and discussed in the main text. [Page 7. Paragraph 6; Page 8. Paragraph 1-2; Page 13. Paragraph 3]*

• Along these lines... I am wondering whether a detergent based definition of protein aggregation could be at least partially the reason of the overrepresentation of DNA binding and nucleolus associated proteins in the "aggregators" category, especially considering that the aggregators largely coincide with the group of heat thermostable proteins found by TPP.

The solubility measure that we used in the definition of aggregators is a ratio between heat and mock shocked samples, which have been both processed the same way. Therefore, any detergent-based biases, for example extractability of DNA-binding proteins, is present in both samples and would be canceled out.

As we show in Figure EV2, proteins can have an insoluble sub-population (i.e. low NP-40 extractability) without any heat shock. The figure also shows that these proteins also include aggregators. As we concluded, protein aggregation is not dependent on the existence of insoluble sub-pool. This can also be stated as NP-40 extractability having no effect on the aggregation propensity.

When it comes to the TPP experiments, the same reasoning applies: the treatment (heat shock) is always compared to a control (mock shock) that has been processed with the same detergent, i.e. all effects stemming from the detergent are cancelled out.

Therefore, the enrichment of nucleolar and DNA-binding proteins in the aggregators from our dataset is a biological consequence of heat shock rather than a technical artifact. We thank the reviewer for pointing out this aspect of possible detergent-bias that we can now clarify in the updated manuscript.

Action taken:

- *We updated the part where we describe the results for GO term enrichment and localization analysis of aggregators (additions in bold):*
 - *“We performed GO term enrichment analysis for the aggregators using all quantified proteins as background. The aggregators were enriched in nuclear proteins involved in DNA binding, chromatin organization and transcription regulator activity (Fig EV3A). The enrichment of nuclear proteins in aggregators was complementarily observed by analyzing protein localization annotations (Fig EV3B); the analysis also indicated that soluble proteins were enriched in cytoplasmic proteins. **Since the aggregators are defined from a comparison between heat and mock shocked samples, all detergent-based or other technical biases related to the lysis conditions would be cancelled out.**”*
[Page 5. Paragraph 5]

- The aggregators are also found to be enriched in hydrophilic and positively charged residues. This, together with the protein complexes membership, are properties that should also differentiate nuclear vs cytosolic proteins. Would you see the same enrichments, if you just did the same tests comparing nuclear and cytosolic proteins?

Interestingly, a very similar question was asked by another reviewer about DNA-binding proteins. We agree that the enrichment could also reflect nuclear proteins.

We have performed the enrichment analysis on nuclear vs cytosolic proteins. We found that from the eight features we compared between aggregators and soluble proteins, six had the same enrichment in nuclear vs cytosolic proteins (i.e. same features enriched in aggregators and nuclear proteins). These included isoelectric point, hydrophobicity, fraction of disordered regions, fraction of random coil, fraction beta sheet and protein complex members. There was no significant difference between nuclear and cytosolic proteins in molecular weight (higher in aggregators). The alpha-helical content was lower for nuclear proteins (no difference between aggregators and soluble proteins).

Action taken:

- *We have added the data of nuclear/cytosolic, as well as the mentioned DNA-binding/no-DNA-binding comparisons in Appendix Figure 3S. The individual comparisons (panels A-F) as well as summarized results (panel G) are shown:*

Appendix Figure 3S - Characteristics of nuclear and DNA-binding proteins.

A: Comparisons of structural and physicochemical features between nuclear and cytosolic proteins shown as combined violin- and boxplots (p-values are for non-parametric Wilcoxon test).

B: Protein complex members in nuclear and cytosolic proteins. Pie charts show the fraction of proteins annotated to be a member of a protein complex. The number of proteins in each segment is indicated. P-value is for Fisher's exact test.

C: Difference in median amino acid composition between nuclear and cytosolic proteins is compared to hydrophobicity (gravy score) for each amino acid. Pearson coefficient (r) with p-value is shown for the correlation analysis.

D-F: As in A-C, expect the comparisons are between DNA-binding proteins (DBP) and all other proteins.

G: Heat map showing the enrichment (or depletion) of molecular features in aggregators (as compared to soluble proteins), DNA-binding proteins (as compared to all other binding proteins) and nuclear proteins (as compared to cytosolic proteins).

Protein was assigned as DNA-binding protein if it contained the GO term for DNA binding (GO:0003677). Protein was assigned as nuclear if it contained any of the following Human Protein Atlas annotations: 'Nucleoli' (GO:0005730), 'Nucleus' (GO:0005634), 'Nucleoplasm' (GO:0005654), 'Nuclear bodies' (GO:0016604), 'Nuclear membrane' (GO:0031965), 'Nuclear speckles' (GO:0016607), or 'Nucleoli fibrillar center' (GO:0001650). Protein was assigned as cytosolic if it contained the Human Protein Atlas annotation 'Cytosol (GO:0005829)' or 'Cytoplasmic bodies (GO:0036464)'. The data is shown for all identified proteins in the MS analysis.

- *A new paragraph in the results section was added to discuss these findings:*

“Many features enriched in aggregators (such as the high amount of disordered regions) could be a result of them being also enriched in nuclear or DNA-binding protein. When analyzing the features presented in Figure 2 for similar enrichment or depletion as observed for aggregators, nuclear and DNA-binding proteins are enriched or depleted in the same features (Appendix Figure 3). However, the molecular weight was similar between nuclear and cytosolic proteins. This suggests that within nuclear proteins, larger proteins tend to aggregate.” [Page 6. Paragraph 2]

- Figure 3: Among the proteins that do not disaggregate after heat shock there are proteins involved in DNA damage. Is their lack of disaggregation compensated by an increase in new protein synthesis? Is this a general trend?

This is a very interesting question. We analyzed the correlation between protein synthesis (as estimated from the 5h time point after heat shock) and disaggregation slope (see figure below). We could not find a correlation between the two. However, we would like to highlight that with such a small sample size, it is challenging to make strong claims about the (lack of) correlation.

Action taken:

- *We added a new figure (Appendix Figure 10S) showing results for correlation analysis between disaggregation and protein synthesis:*

- *We added a description of the results in the main text:*
 - *“Since some proteins that aggregated in heat shock were not disaggregated (Fig 3C), they possibly were replaced by upregulated protein synthesis. We could not find correlation between disaggregation and protein synthesis (Appendix Figure 10). However, it should be noted that the sample size in the analysis is relatively low due to the lower coverage of the newly synthesized protein fraction.” [Page 10. Paragraph 1]*

- The dynamic SILAC experiment done to analyse newly synthesized proteins reveals that the most upregulated proteins include many heat shock proteins. This is in line with what has been observed before in other whole proteome analysis of the heat shock response (including time courses). Perhaps the authors should highlight more what we learn from this new dynamic SILAC experiment in the context of heat shock.

The main reason to use dynamic SILAC was to filter out protein synthesis to allow for more accurate measurement of disaggregation. However, as we state in the manuscript, “the dynamic SILAC approach allowed to analyze also newly synthesized proteins”, although with lower coverage.

We agree that the upregulation of heat shock proteins is in line with what has been observed before. In addition, the global stall of protein synthesis is also observed before (and is now stated in the updated manuscript). We think that both of these observations have been well-established a long time ago in the field and we certainly do not claim that they are our new findings. However, we think that they serve as proof-of-concept for the dSILAC experiment.

We show that the proteins aggregating in the newly synthesized fraction are predominantly the same as in the pre-existing fraction. Related to a comment by another reviewer, now we also quantified the aggregation in both SILAC fractions and can also show that proteins aggregate to very similar extent

upon heat shock in both of them. We think this is an interesting finding since newly synthesized proteins are generally thought to be less or more prone for aggregation.

In the manuscript we compare the protein synthesis levels with previously reported mRNA-level regulation upon heat shock. We can show that the mRNA-levels measured after heat shock do not correlate with protein synthesis at five hours after recovery (when the upregulation is evident). However, heat shock proteins are co-regulated on both levels. We think that showing this for human cells [as similar finding has been made with yeast (Muhlhofer et al. 2019, DOI: 10.1016/j.celrep.2019.11.109)] is a new information.

To the best of our knowledge, we have provided the first dataset of protein synthesis regulation after heat shock in human cells that can be quantified for more than a thousand proteins. Based on the suggestion of another reviewer, we have now added in the manuscript discussion about co-regulation of disaggregase components after heat shock. We would like to highlight that the hypothesis for this co-regulation stemmed from our dataset and as such gives now an example of what could be learned from the experiment.

Action taken:

- *We added a statement that the stall in protein synthesis is in line with previous studies. [Page 9. Paragraph 4]*
- *Results from comparing aggregation of newly synthesized and pre-existing proteins are discussed in the text:*
 - *“In addition, the solubility change upon heat shock was very similar for aggregators in both SILAC fractions (Appendix Figure 8SA). The small differences between both protein populations did not correlate with disordered regions (Appendix Figure 8SB).” [Page 9. Paragraph 5]*
- *Discussion related to co-regulation of disaggregase components is added to the text:*
 - *“Interestingly, both highly upregulated Hsp40s (DNAJB1 and DNAJB4) belonged to class B of Hsp40s which are involved in protein disaggregation [27]. Although other Hsp40s were also upregulated (Appendix Figure 9S), this suggested that the response to heat shock involved an increase in disaggregation capacity by upregulated protein synthesis.” [Page 9. Paragraph 6]*
 - *“We noted that the most upregulated heat shock proteins could, in theory, form a Hsp70-based disaggregase if assuming that the components would be co-expressed. These proteins include a Hsp70 (HSPA1A| HSPA1B) with a nucleotide exchange factor (HSPH1) and two Hsp40 proteins (DNAJB1 and DNAJB4). To speculate further, the two*

class B Hsp40s could reflect an adjustment towards clearance of larger aggregates, as it has been suggested for class B in contrast to class A (which are involved in clearing smaller size aggregates) [27]. However, it should be noted that other Hsp40s are upregulated too, for example DNAJC8, that has no known link to disaggregation. Further studies would help to understand the co-expression of disaggregase components and the possible adjustment of its composition to reflect the aggregation load.” [Page 14. Paragraph 2]

- Dynamic SILAC experiment, figure 4C: here the increase in protein neo-synthesis was compared with published mRNA variations upon yeast stress. The authors here write that: "other proteins were upregulated only at the protein level (Figure 4G), suggesting that their upregulation is translation rather than transcription driven". Considering that the majority of upregulated proteins of figure 4G are also upregulated at the mRNA level (7/11), I would mitigate this statement.

We appreciate the reviewer’s point that majority of upregulated proteins have mRNA and protein level upregulation and this could be the only conclusion to be made. However, we think that the complete and clear separation between heat shock proteins and non-heat shock proteins is meaningful and should be addressed.

Action taken:

- *We modified the statement in the manuscript to include both, the conclusion made by the reviewer as well as our reasoning behind the original statement. We think that the statement now includes an unbiased description of the data followed by our interpretation that is marked as our subjective conclusion (deletions with strikethrough and additions in bold):*

*“~~Interestingly, f~~ From our ten most upregulated proteins, **the majority of them** ~~only heat shock proteins~~ were also upregulated at the mRNA level; ~~other proteins were upregulated only at the protein level~~-(Fig 4F). **We noticed that the upregulated heat shock proteins were the only ones with upregulation also on mRNA level; non-heat shock proteins were upregulated only on the protein level,** suggesting that their upregulation ~~iswas not driven by translation rather than mRNA levels transcription driven~~. Similar findings have been made with yeast [53].” [Page 10. Paragraph 2]*

- I find interesting that the soluble fraction of many "aggregators" proteins become more thermally stable. Perhaps this suggests that there are subpopulations of proteins with different melting temperature. What would be the outcome of this experiment if the heavy labelled (newly synthesized proteins) are analysed ?

We appreciate the interest showed towards one of our main findings. The suggestion that there would be a sub-population of proteins with different melting temperature is exactly how we think the results

should be interpreted; in a sense, increase in thermal stability means higher melting temperature. The power of 2D-TPP is that the thermal stability is not measured as melting temperature and, therefore, is not bound to the assumptions required for typical melting curve.

It would be very interesting to analyze the stability changes of newly synthesized proteins. However, we performed the 2D-TPP experiment without the dynamic SILAC and cannot offer these data in the manuscript. As indicated for the aggregation/disaggregation dataset, using dSILAC results in much lower coverage for the newly synthesized proteins due to lower signal intensity. In 2D-TPP, this lower intensity would probably be undetectable in the higher temperature range where most proteins would have aggregated. Therefore, the stability data for newly synthesized proteins would probably be possible for only a few highly abundant proteins using dSILAC. For a proper experiment, one would need a different protein labelling approach (such as used in Savitski et al. 2018, DOI: 10.1016/j.cell.2018.02.030) combined with peptide labelling method that have higher multiplexing capacity than TMT11. We hope that in the near future these technical challenges are met.

If the 2D-TPP would be done on newly synthesized proteins, we speculate that many proteins would have lower thermal stability. The reasons behind this could be incomplete folding (that would be impacted more by increased demand for chaperones upon heat shock), for example. On the other hand, when we analyzed the aggregation of newly synthesized proteins (Figure 4E in the first version of the manuscript), the same proteins tend to aggregate than in the pre-existing protein fraction.

Action taken:

- We added discussion related to analyzing newly synthesized proteins with 2D-TPP:
 - “While the 2D-TPP assay discussed here reflects the response of mainly pre-existing proteins, it would be interesting to develop the method further to analyze heat-induced stability changes of newly synthesized proteins. For example, one could speculate that the stability of newly synthesized proteins could be affected by incomplete folding or chaperone binding. Extending the analysis by using chaperone inhibitors could help to understand these processes.” [Page 14. Paragraph 3]

- In the thermal stability experiment I did not quite understand why the heat shock-induced aggregation is adjusted with changes measured at 37 and 37.8 degrees, if the heat shock was done at 44 degrees. The reference to the material and methods does not address this.

To estimate the solubility change caused by the heat shock-induced aggregation, one would compare heat shocked samples to mock shocked samples. This is exactly what is done by measuring the change at 37.0 and 37.8°C - we assume that nothing has aggregated at these temperatures. If, on the other hand, the difference would be estimated at 44.0°C - the heat shock temperature - one would not expect to see any difference between the heat and mock shocked samples. This is because they both experienced the

44.0°C temperature and proteins would aggregate to approximately same extent also in the mock shocked sample.

Action taken:

- We updated the materials and methods section regarding this topic (additions in bold):
*“ . . . the average difference in the first two temperature points (**37.0°C and 37.8°C**) were subtracted from all temperature points. **When correcting for the aggregation, we assume that no aggregation has taken place in the mock shocked sample in these temperatures.**”*
[Page 26. Paragraph 4]

MINOR POINTS

- The order of the panels of the figures is not always consistent with the text. Please correct.

We apologize for the inconsistency and thank the reviewer for pointing it out.

Action taken:

- We have made thorough re-arrangements to Figures EV2 and EV4:
 - Panel B in the original Figure EV2 that shows protein intensities in SDS-lysed samples from all time points and both SILAC fractions has been separated to three panels
 - Values after heat shock (light SILAC) is now Figure EV2C
 - Values during recovery (light SILAC) is now Figure EV4A
 - The new panel is described in the text:
“The total protein intensity remained constant during the recovery (Fig EV4A). This indicated that the increased intensity in the soluble fraction during recovery for aggregators stemmed from increasing solubility. In addition, this also showed that heat shock did not induce protein degradation.” [Page 7. Paragraph 4]
 - Values after heat hock and during recovery (heavy SILAC) is now Figure EV4D
 - Protein complex analysis of disaggregation in Figure EV4A is moved to figure EV4C
 - Protein complex analysis of aggregation in Figure EV4B is moved to Figure EV2G
 - Figure EV2E showing comparing disaggregation slopes and the presence of insoluble protein fraction is moved to Figure EV4B
- As a result of the re-arrangements, Figure EV2 now has the focus on aggregation results and Figure EV4 on the recovery phase results (i.e. disaggregation and protein synthesis).
 - Title of Figure EV2 changed to “Protein abundance, total solubility and aggregation upon heat shock.”

- Title of Figure EV4 changed to “Disaggregation and protein synthesis during recovery from heat shock”
- In the text, two paragraphs describing results presented in Figure EV3 (GO term enrichment, localization analysis and proteins related to stress granules) are now re-located so that the results in Figure 2 and Figure EV2 are described first. As a result, the molecular features of aggregators are described before functional analysis is discussed. We think that this change in the order of the way results are described does not impact the flow of the text nor change any of the conclusions made from them. [Page 4. Paragraph 3-4; Page 5. Paragraph 5; Page 6. Paragraph 1]

- The authors mentioned that they performed initial control experiments to ensure that the intracellular pool of arginines and lysines was consumed from the light SILAC medium before heat shock. These experiments could be reported as supplementary information.

*Unfortunately, we do not have such data. We assume that this misunderstanding might come from the way we wrote about it (“This period prior to heat treatment **ensured** that the intracellular pool of arginine and lysine from light SILAC medium was consumed”).*

With the dSILAC, it would be very challenging to measure when the light label stops accumulating to newly synthesized proteins. This is because the whole proteome has light labelled arginine and lysine at the moment of medium switch. The residual light amino acids that would be added to the newly synthesized proteins would be such a small fraction of all light labelled proteins that the signal would be undetectable. However, we assume that since after 90 minutes from the medium switch we can robustly detect over 1000 proteins with the heavy label, the residual light amino acids are consumed or are present in such a small quantity that they can be neglected.

Action taken:

- We updated the section where we discuss the timing of the medium switch (deletions with strikethrough and additions in bold):
 - “Dynamic SILAC labelling [29, 30] was used to distinguish between pre-existing proteins and newly synthesized proteins (Fig 1A). K562 human leukemia cells were cultured in light SILAC medium. The medium was switched to heavy SILAC 90 minutes before heat treatment. **We assume that during** ~~this period prior to heat treatment, ensured that~~ the intracellular pool of arginine and lysine from light SILAC medium **would be** ~~was~~ consumed. **That would** ~~which~~ allowed a more accurate quantification of pre-existing proteins **since the signal from newly synthesized proteins can be filtered out.**” [Page 3. Paragraph 2]

- Figure EV2B: If the purpose of this figure is to show that the total protein amount remains constant

over 10 minutes of heat shock, why are the correlation plots relative to the recovery time points also shown?

We refer to different parts of Figure EV2B in the text and, as the reviewer points out, this can be confusing.

Action taken:

- *Related also to a previous point from the reviewer about the order of panels, we have split Figure EV2B to three panels:*
 - *Values after heat shock (light SILAC) is now Figure EV2C*
 - *Values during recovery (light SILAC) is now Figure EV4A*
 - *Values after heat hock and during recovery (heavy SILAC) is now Figure EV4D*

- The GO enrichment analysis for the aggregators should specify the statistics applied and the statistical significance cut off applied.

We apologize for the missing information.

Action taken:

- *A statement indicating the statistical test and significance cut-off has been added to materials and methods:*
 - *“Enrichment analysis was based on hypergeometric test with a cut-off of 0.05 for Benjamini-Hochberg adjusted p-value.” [Page 27. Paragraph 8]*
- *The legend of Figure EV3 was updated (additions in bold):*
 - *“Bar plot showing ten most enriched (lowest **Benjamini-Hochberg adjusted p-value in hypergeometric test**) terms from each GO domain.”*

Reviewer #3:

The manuscript by Määttä and coworkers analyzed the aggregation and disaggregation of human proteins under heat stress using mass spectrometry-based proteomics in combination with dynamic SILAC and TMT multiplexing. They first defined aggregating proteins by their absence in the soluble fraction upon mild heat stress compared to control. They found the proteins in the analyzed aggregates to have on average distinct physicochemical properties (higher disorder, higher molecular mass, higher

iP and more hydrophilic AA content) compared to non-aggregating proteins and that proteins in the aggregates are more likely part of large multi-protein complexes. Second, they report that aggregated proteins disaggregated over time, with proteins with higher disorder showing faster aggregation and disaggregation rates. Finally, the authors provide 2D-TPP data to show that a sub-population of aggregating proteins remains soluble after heat shock and shows increased thermal stability afterwards.

Overall, this is an interesting dataset based on state-of-the-art mass spec technology. While the presented data analysis and the concepts discussed are interesting, there are a number of points the authors need to address before the paper could become suitable for publication. First, there is no independent experimental validation of the mass spec data. I would have expected to see at least some immunofluorescence data on the subcellular distribution of proteins before and after heat shock. This would also indicate which type of "aggregates" actually mediates the solubility changes they observed. Second, data analysis and discussion misses important aspects related to disordered proteins and their role in protein-protein interactions and protein toxicity. Third, the authors fail to analyze/discuss the possibility that the heat-shock induced thermal stabilization of proteins could just reflect different cellular pools of the same protein. Also, the data is currently inaccessible for reviewers and no supplemental tables are provided. Therefore, while I expect the data to be of high quality given the descriptions provided and the excellent track record of the lab, there is no way for me to assess this.

Specific points:

1. There is no independent experimental validation of the data. I think immunofluorescence microscopy with antibodies against a number of aggregating proteins could be used to (i) validate the MS data and (ii) show how and where the proteins aggregate and disaggregate.

We agree that this would be an important experiment to conduct. We performed immunofluorescence analysis for three aggregators (HELLS, BRD4 and TARDBP) and one soluble protein (HDAC1) to analyze the localization of these proteins upon heat shock and during recovery. From the aggregators, BRD4 and TARDBP disaggregated in the mass spectrometry experiment, while HELLS stayed aggregated for the used recovery time of five hours.

In the microscopy experiment, it was found that (1) all proteins were located at the nucleus confirming the Human Protein Atlas localization annotation, (2) a fraction of these proteins translocated to cytoplasm upon heat shock where especially HELLS formed foci, and (3) the cytoplasmic signal returned to levels comparable to control after five hours of recovery except for HELLS. On the other hand, non-aggregating HDAC1 showed unchanged nuclear localization upon heat shock and during recovery. Therefore, the cytoplasmic translocation and (foci formation for HELLS) upon heat shock as well as the reverse during recovery coincide with the observed changes in solubility detected in the mass spectrometry experiment.

We also analyzed different times for the heat shock. We found that HELLS had peculiar localization depending on the duration of the heat shock. After ten minutes, a small sub-population translocated to cytoplasm and formed foci (as discussed above). However, if the heat shock was prolonged to 120

minutes, most signal came from the foci formed near the nuclear membranes. On the other hand, BRD4 formed larger cytoplasmic foci at prolonged heat shock while TARDBP started to form nuclear foci.

Action taken:

- Immunofluorescence images added to Figure 3D and Appendix Figure 6S

- Figure 3D:

○ Appendix Figure 6S:

● Results discussed in the text:

- “We validated the aggregation and disaggregation propensities of a few proteins from our dataset using immunofluorescence. HDAC1, a non-aggregating protein, showed nuclear localization in control conditions and remained unchanged upon heat shock and during recovery (Fig 3D, Appendix Figure 6SA). On the other hand, aggregators HELLS, BRD4 and TARDBP (all localized in the nucleus) showed increased intensity in the cytoplasm upon heat shock (Fig 3D, Appendix Figure 6SA). The increased cytoplasmic signal was strongest for HELLS (the most aggregating protein in the mass spectrometry

experiment) while the cytoplasmic signal was weaker for BRD4 and TARDBP. HELLS seemed to form foci in the cytoplasm. Interestingly, prolonged heat shock caused the cytoplasmic and nuclear HELLS to localize and form foci at nuclear membranes (Appendix Figure 6SB). The cytoplasmic BRD4 formed bigger foci during prolonged heat shock while TARDBP formed nuclear foci (Appendix Figure 6SB).” [Page 7. Paragraph 6; Page 8. Paragraph 1]

- *“After five hours of recovery, the cytoplasmic signal of BRD4 and TARDBP was comparable to control level, while the cytoplasmic HELLS remained in foci (Fig 3D, Appendix Figure 6SA). These findings corroborated the observations from the mass spectrometry experiment, where BRD4 and TARDBP disaggregated while HELLS remained aggregated in the insoluble fraction during the recovery. The solubility changes of nuclear proteins observed with mass spectrometry upon heat shock and during recovery coincide with protein transport to cytoplasm and foci formation.” [Page 8. Paragraph 2]*
- *“The formation of cytoplasmic foci that coincide with the aggregation detected in mass spectrometry experiment suggests that nuclear proteins do not aggregate in the nucleus, but rather translocate and aggregate in cytoplasm. However, the link between foci formation and loss of solubility is not trivial. Small aggregates that are beyond the detection limit of microscopy could contribute to a large fraction of the solubility decrease observed for aggregators [16]. In addition, majority of proteins related to cytoplasmic stress granules remained soluble (Figure EV3C). Therefore, we would hesitate to conclude that loss of solubility could be explained by cytoplasmic foci formation. Although some evidence connects foci formation and solubility change in mass spectrometry-based assays [12], more focused studies would be required to investigate the issue.” [Page 13. Paragraph 3]*

2. There is considerable data on the role of intrinsic protein disorder in dosage sensitivity (Vavouri et al., Cell, 2009; DOI: 10.1016/j.cell.2009.04.029), liquid-liquid phase separation (Bolognesi et al., Cell Rep., 2016, DOI: 10.1016/j.celrep.2016.05.076), protein complex formation (McShane et al., Cell, 2016, DOI: 10.1016/j.cell.2016.09.015) and aggregation during aneuploidy (Brennan et al., Genes Dev, 2019, doi: 10.1101/gad.327494.119). My main critique is that the findings should be analyzed and discussed in the light of these previous findings. For example, since disorder appears to be a key determinant of dosage sensitivity (Vavouri et al.) it would make sense to sequester them in aggregates upon overexpression, which indeed has been shown (Brennan et al.). Also, given the potentially toxic nature of disordered proteins, it would make sense to make sure that they are not made in excess relative to their interaction partners, which was also shown (McShane et al.). Heat shock might liberate disordered proteins from existing complexes or disrupt de novo assembly of complexes from newly synthesized proteins, which in both cases could lead to cellular overabundance of disordered proteins. In this context, it could also be interesting to look at differences in the aggregation propensity of preexisting and newly synthesized proteins, which should be possible in the dynamic SILAC dataset.

We are very thankful for the reviewer to extend the discussion about disordered regions. The potential toxicity of disordered regions was introduced in the discussion and we would like to extend it with the references given by the reviewer and make the implication of the possible protective aggregation more explicit.

As suggested by the reviewer, we compared to aggregation propensity between pre-existing and newly synthesized proteins (Appendix Figure 8SA shown below). Generally, the solubility of aggregators has the same trends in both SILAC fractions. We then analyzed the possible difference by comparing the solubility difference between heavy and light to the amount of disorder (Appendix Figure 8SB shown below). We could not find differences in the aggregation propensity in the different SILAC fraction that could be related to disordered regions. In the context discussed by the reviewer, these results suggest that proteins with disordered regions have the same aggregation propensity in pre-existing fraction (possibly released from assembled complexed) and newly synthesized fraction (not assembled to complexes).

Action taken:

- We added a new figure (Appendix Figure 8S) that shows the comparison between the solubility of pre-existing and newly synthesized proteins upon heat shock (A), as well as comparison of the solubility difference and fraction disorder for aggregators (B):

- We describe the results in the main text:
 - ““In addition, the solubility change upon heat shock was very similar for aggregators in both SILAC fractions (Appendix Figure 8SA). The small differences between both protein populations did not correlate with disordered regions (Appendix Figure 8SB).” [Page 9. Paragraph 5]

- *We added the following text after we discussed potentially toxic nature of disordered regions:*

“In addition, the involvement of disordered regions in dosage sensitivity has been demonstrated [Vavouri et al.; Bolognesi et al.]. Indeed, overexpressed disordered proteins from aneuploidic chromosomes are sequestered to aggregates [Brennan et al.], degraded faster when present in complexes with super-stoichiometric amounts [McShane et al.] and form toxic cytoplasmic granules when present with high concentration [Bolognesi et al.]. Therefore, aggregation of disordered proteins upon heat shock could protect cells from their potentially toxic effects.” [Page 12. Paragraph 5]

3. Related to the observed changes in thermal stability after heat shock, I am wondering how much of this simply reflects different protein pools. For example, assuming that the free fraction of DNA-binding proteins is more sensitive to heat-induced aggregation than the DNA-bound fraction, we would expect that applying the heat shock depletes the free fraction and selects the DNA-bound fraction, which will then appear stabilised in the subsequent TPP experiment. However, the thermal stability of the proteins is actually not changing. Rather, TPP on mock-shocked and heat-shocked cells involves looking at different subpopulations of protein molecules which already had different thermal stability before the heat shock. This also relates to the discussion section: The authors write "it is tempting to speculate that chromatin binding could also have protein quality aspects by stabilizing and sequestering unstable DNA-binding proteins during proteotoxic stress". The more parsimonious explanation is that the DNA-bound fraction is (i) generally more thermally stable and (ii) less likely to aggregate upon heat shock (note that these two points are probably also biophysically related to each other). Applying the heat shock merely selects for the DNA-bound fraction of a protein, which is why it looks more thermally stable. There is no need to speculate about a possible role of chromatin in protein quality control. More generally, proteins can exist in different "states" in the cell (different interaction partners, proteoforms, PTMs) that have different thermostability and correlated differences in aggregation propensity.

We completely agree with the thorough reasoning the reviewer has provided. It is true that we cannot state how much of the stability signal is due to stable sub-population present in the cells before the heat shock and what is the contribution of heat-induced stability. In this light, we also think that the speculative quality control role for chromatin might be too strong claim to make from the data.

Action taken:

- *We have removed the following text where we speculated about chromatin as a quality control site and did a minor adjustment to the sentence that follows (removed parts with strikethrough and additions in bold):*

- *“Aggregation in the nucleus could mean several different things. For example, during stress, proteins have been reported to enter the nucleoli [69] and increase abundance at chromatin [70]. ~~Although the nucleolus is actually proposed to act as a protein quality compartment [69], it is tempting to speculate that chromatin binding could also have protein quality aspects by stabilizing and sequestering unstable DNA-binding proteins during proteotoxic stress.~~ **These can** could be linked to our results of increased thermal stability for soluble sub-pools of aggregators that did not aggregate in heat shock (Fig 5C); the most strongly thermally stabilized proteins were indeed zinc-finger containing DNA-binding proteins (Fig 5C).” [Page 12. Paragraph 4]*
- *We have added a statement regarding the stability effect possibly stemming from more stable sub-populations present without heat shock (removed parts with strikethrough and additions in bold):*
 - *“The thermal stabilization of soluble remnants of aggregating proteins could reflect an instant post-translational mechanism of induced thermotolerance. **Although, it should be noted that we cannot conclude how much of the increased stability signal is stemming from more stable protein sub-population that was already present in the sample before the heat shock and how much of the signal reflects heat shock-induced stability changes. For the heat shock-induced stabilization, ~~The~~ the 2D-TPP assay, as we applied it here, can be viewed as a way to measure instantly gained thermotolerance without transcriptional or translational regulation.**” [Page 14. Paragraph 3]*
- *Related to previous modification and other comments, we decided to remove the following statement from the abstract:*
 - *“Our results provide a rich resource of heat stress-related protein solubility data, propose novel roles for intrinsically disordered regions in protein quality control and reveal a protection mechanism to repress protein aggregation in heat stress.” [Page 1. Abstract]*

4. The proteomic data has to be made available to reviewers (both the raw files and processed files).

We apologize that this was not made available to reviewers. We had uploaded the raw data to PRIDE at the time of submission, and provided the credentials to access the raw data during our initial submission.

Action taken:

- *The PRIDE credentials are as follows:*
 - *username: **reviewer32979@ebi.ac.uk***
 - *password: **zQdAO1U***

5. While the light labeled proteins are in steady state condition, the heavy channel contains newly synthesized proteins (after 90 mins). This makes the interpretation of the difference in solubilities between subunits within a complex more complicated. The authors cannot distinguish between free non-assembled subunits and assembled subunits. At 90 mins the assembly of many complexes will not be finished. Thus, the signal is a mixture of both protein pools (unassembled and assembled). Proteins incorporated into a complex likely have different melting curves compared to their free counterparts (Tan et al. (2018), Science).

We completely agree with the reasoning. However, here we also took advantage of the dSILAC labelling. As stated in the figure legend of Figure 2, the results for the protein complexes is for the pre-existing proteins (light) only. Therefore, the signal is for assembled complexes and not a mixture of assembled and unassembled complexes. After processing the comments from all reviewers, we have appreciated that this aspect has not always been clear in the manuscript.

Action taken:

- *The SILAC fraction under analysis has been indicated in figures presenting results for protein complexes. [Figure 2, Figure EV2G and EV4C in the updated manuscript]*

6. Aneuploidy has recently been shown to increase protein aggregation in yeast (Brennan et al. 2019). Since K562 are aneuploid the authors may want to reanalyze their data in this context (enrichment for proteins encoded in genomically amplified regions). Aneuploidy might help explain some of the observed aggregation before heat stress.

We thank the reviewer for this excellent suggestion and have explored this aspect further. We have now analyzed the frequency of aggregators and soluble proteins in all chromosomes. We did not find any evidence for over- or under-representation of aggregators in any of the chromosomes.

Action taken:

- *We have added the results as a supplementary table in the updated manuscript. (Appendix Table 1S):*

Appendix Table 1S - Frequencies of aggregators and soluble proteins in different chromosomes.

Chromosome	Aggregators	Soluble	p-value*	Adjusted p-value#
1	16	223	0.517	1.000
2	13	139	0.184	1.000
3	8	125	0.597	1.000
4	4	71	0.614	1.000
5	7	99	0.471	1.000
6	8	104	0.390	1.000
7	11	112	0.157	1.000
8	4	73	0.638	1.000
9	4	65	0.539	1.000
10	4	77	0.682	1.000
11	9	122	0.442	1.000
12	9	111	0.337	1.000
13	1	38	0.770	1.000
14	7	70	0.173	1.000
15	4	70	0.602	1.000
16	2	93	0.967	1.000
17	10	130	0.397	1.000
18	2	27	0.334	1.000
19	12	125	0.168	1.000
20	6	57	0.153	1.000
21	3	14	0.028	0.667
22	6	48	0.083	1.000
X	7	82	0.289	1.000

* P-values were obtained using a Monte Carlo sampling approach (B = 1000) using a binomial distribution with probabilities obtained from the overall frequencies of soluble and aggregating proteins. It was then tested whether the numbers of observed soluble and aggregating proteins from each individual chromosome were significantly different from the overall trend by comparison to the sampling results.

P-values are Benjamini-Hochberg adjusted for multiple hypothesis testing.

- *We added a description of the results to the main text:*

“Chromosome duplications can lead to gene overexpression. It has been shown that the protein overproduction is counteracted by aggregation [Brennan et al. 2019]. As K562 cells contain aneuploidic chromosomes, we analyzed the frequency of aggregators in each chromosome to test whether aggregators would be over-represented in duplicated chromosomes. We found no difference in the occurrence of aggregators or soluble proteins in any of the 23 chromosome (Appendix Table 1S) suggesting that protein overproduction does not contribute to aggregation propensity upon heat shock.” [Page 6. Paragraph 3]

7. "Directly after heat shock, the abundance of 300 proteins (<7% of quantifiable proteome) decreased significantly (Benjamini-Hochberg adjusted p-value < 0.05 in LIMMA analysis and fold change < 2/3) in the soluble fraction when compared to mock shocked sample (Fig 2A)." It is not clear to me off this statement refers to the preexisting (light) proteins or the newly synthesised (heavy) ones. This should be clarified.

The statement refers to the pre-existing (light) fraction. This information was presented in the figure legend. We agree that this should be clarified further since another reviewer had the same comment.

Action taken:

- *The SILAC fraction under analysis is annotated in Figure 2.*
- *A statement about the used SILAC fraction is added before we describe the aggregation results:*

“The protein aggregation was analyzed from the pre-existing (light) fraction.” [Page 3. Paragraph 6]

8. Figure 2I: It took me a while to understand this figure because of the missing column labels. The rows are clearly labeled with the subunits but the columns are not. I assume the columns also represent complex subunits. This should be indicated for clarification.

We thank the reviewer for pointing out the un-clarity in the figure which has been fixed.

Action taken:

- *Columns labelled in Fig 2I.*

9. "Together, these results indicate that the disaggregation is the preferred method to deal with aggregates." The word "preferred" implies that disaggregations is somehow advantageous, which cannot be concluded from the data in figure 3 B and C.

We completely agree with the reviewer. Our aim was not to claim that the data could show any advantage of disaggregation over other ways to handle aggregates. We changed the wording from “preferred method” to “main strategy” as this term has been used previously in this context (Mogk et al. 2018, DOI: 10.1016/j.molcel.2018.01.004).

Action taken:

- *“Preferred method” change to “main strategy”. [Page 7. Paragraph 3]*

10. "we found a significant (Benjamini-Hochberg adjusted p-value < 0.05) negative correlation between disaggregation slope and the loss of solubility after heat shock (Fig 3F; $p = 1.23 \times 10^{-13}$; $r = -0.42$). In other words, the more a protein aggregated, the faster it disaggregated during the recovery." I am wondering if there might be a technical reason for this observation. In essence, the observation is that some proteins show larger changes (during aggregation and disaggregation) and others smaller ones. Could this be related to systematically different degrees of TMT ratio compression?

This is an excellent and valid point. We have analyzed this aspect by comparing signal to interference values between aggregators and proteins that stay soluble. Proteins measured with low signal to interference values are more prone for TMT ratio compression.

We could not find any evidence that aggregators in general (Appendix Figure 7SA shown below), the solubility after heat shock (Appendix Figure 7SB shown below) nor the disaggregation slopes (Appendix Figure 8SC shown below) would be linked to low signal to interference values.

In addition, the data processing tool that we used (isobarQuant: Franken et al. 2015, DOI: 10.1038/nprot.2015.101) has an in-built algorithm (Savitski et al. 2013, DOI: 10.1021/pr400098r) that strongly decrease the possible ratio compression stemming from peptide co-fragmentation.

Therefore, the correlation between aggregation and disaggregation is most likely not because of TMT ratio compression

Action taken:

- *We added a new figure (Appendix Figure 7S) where the results discussed above are shown:*

- *We discuss ratio compression and our results in the main text:*
 - *“The correlation between solubility and disaggregation slopes could be a result of ratio compression during TMT quantification, which may stem from peptide co-fragmentation [56]. We therefore analyzed signal to interference values (which are*

lower for proteins prone for ratio compression) in the context of aggregation and disaggregation. We could not find any signs of lowered signal to interference values for aggregators (Appendix Figure 7SA). In addition, no correlation was observed between signal to interference and solubility (Appendix Figure 7SB) nor disaggregation slope (Appendix Figure 7SC). Therefore, it is unlikely that ratio compression played a role in the observed correlation between solubility and disaggregation.” [Page 8. Paragraph 4]

11. Certain subgroups of proteins (transcription factors, DNA-binding proteins, perhaps ribosomal proteins) are overrepresented in the aggregated proteins. This begs the question whether molecular features associated with aggregation/disaggregation (disorder, amino acid composition) are truly features of aggregation/disaggregation or merely features of these protein subgroups. I realise this is a chicken and egg question that is hard to answer. But it might be possible to take a look at two distinct subgroups of aggregators (such as DNA-binders versus non-DNA binders) in order to see if they share similar molecular features or not.

This is a very interesting idea. Another reviewer has suggested the same analysis on the context of nuclear proteins. We found that most of the molecular features enriched in aggregators are also enriched in DNA-binding proteins. The exception is fraction of predicted alpha helical structure.

Action taken:

- *The DNA-binders versus non-DNA-binders analysis is added as Appendix Figure 3S together with the same comparison of nuclear vs cytosolic proteins:*

Appendix Figure 3S - Characteristics of nuclear and DNA-binding proteins.

A: Comparisons of structural and physicochemical features between nuclear and cytosolic proteins shown as combined violin- and boxplots (p-values are for non-parametric Wilcoxon test).

B: Protein complex members in nuclear and cytosolic proteins. Pie charts show the fraction of proteins annotated to be a member of a protein complex. The number of proteins in each segment is indicated. P-value is for Fisher's exact test.

C: Difference in median amino acid composition between nuclear and cytosolic proteins is compared to hydrophobicity (gravy score) for each amino acid. Pearson coefficient (r) with p-value is shown for the correlation analysis.

D-F: As in A-C, expect the comparisons are between DNA-binding proteins (DBP) and all other proteins.

G: Heat map showing the enrichment (or depletion) of molecular features in aggregators (as compared to soluble proteins), DNA-binding proteins (as compared to all other binding proteins) and nuclear proteins (as compared to cytosolic proteins).

Protein was assigned as DNA-binding protein if it contained the GO term for DNA binding (GO:0003677). Protein was assigned as nuclear if it contained any of the following Human Protein Atlas annotations: 'Nucleoli' (GO:0005730), 'Nucleus' (GO:0005634), 'Nucleoplasm' (GO:0005654), 'Nuclear bodies' (GO:0016604), 'Nuclear membrane' (GO:0031965), 'Nuclear speckles' (GO:0016607), or 'Nucleoli fibrillar center' (GO:0001650). Protein was assigned as cytosolic if it contained the Human Protein Atlas annotation 'Cytosol (GO:0005829)' or 'Cytoplasmic bodies (GO:0036464)'. The data is shown for all identified proteins in the MS analysis.

- *The results are discussed in a new paragraph with the similar results found for nuclear versus cytosolic proteins:*
 - *"Many features enriched in aggregators (such as the high amount of disordered regions) could be a result of them being also enriched in nuclear or DNA-binding protein. When analyzing the features presented in Figure 2 for similar enrichment or depletion as observed for aggregators, nuclear and DNA-binding proteins are enriched or depleted in the same features (Appendix Figure 3). However, the molecular weight was similar between nuclear and cytosolic proteins. This suggests that within nuclear proteins, larger proteins tend to aggregate." [Page 6. Paragraph 2]*

12. "Within a protein complex (n = 32), aggregators had more similar disaggregation profiles when compared to scrambled complexes (n = 10000) containing the same aggregators randomly re-distributed (Fig EV4A; p = 0.048)." It should be stressed that this signal is quite weak. Also, I assume that taking more than 10000 scrambled complexes would make the p-value more significant. Hence, the functional significance of the nominally significant p-value is not clear to me.

We agree that the strength of the signal should be stated. The 10 000 scrambled complexes should indeed make the p-value more significant. However, we think that the low number of real complexes would impact the p-value as well. Therefore, we think that the nominally significant p-value is indicative of a trend in the data.

Being fully aware of the relatively weak signal, we described the result in the original manuscript by stating that

“aggregators had more similar disaggregation profiles when compared to scrambled complexes”

and

“. . . suggesting that aggregators in complexes aggregate to different extent but can disaggregate similarly.”

We think that these statements are quite neutral and are not trying to make any strong claims about the data.

Action taken:

- *We added a notion about the weak signal in the text (additions in bold):*
 - *“Within a protein complex (n = 32), aggregators had **a weak trend for** more similar disaggregation profiles when compared to scrambled complexes. . .” [Page 8. Paragraph 5]*

13. "Interestingly, from our ten most upregulated proteins, only heat shock proteins were also upregulated on mRNA level; other proteins were upregulated only at the protein level (Fig 4G) suggesting that their upregulation is translation rather than transcription driven." Alternatively, the observed protein-level up-regulation could reflect decreased protein degradation. No direct evidence for translational control is presented.

We agree with the reviewer that the decreased protein degradation could explain the upregulation and that there is no direct evidence for translational control.

Action taken:

- *We have updated our interpretation of the results (deletions in strikethrough and additions in bold):*

*“. . . suggesting that their upregulation is ~~translation rather than transcription driven~~ **not driven by mRNA levels.**” [Page 10. Paragraph 2]*

14. "Therefore, we speculate whether disordered regions could serve other functions not related to sequestering proteins to aggregates. Disordered regions could offer, for example, shielding for

potentially toxic protein-protein interactions, such as seeding and formation of amyloid fibers." As mentioned in point 1, disorder itself appears to be toxic, and cells generally tend *not* to overproduce disordered proteins that are part of multiprotein complexes. Therefore, the idea that disordered regions could somehow "shield" toxic protein-protein interactions does not sound plausible to me.

The part of the discussion that the reviewer is pointing includes maybe the most speculative discussion we included in the manuscript. After also considering a previous point made by the reviewer about disordered regions we decided to leave out this speculation.

Action taken:

- *We have removed the following text from the discussion:*
 - “Disordered regions could offer, for example, shielding for potentially toxic protein-protein interactions, such as seeding and formation of amyloid fibers. This kind of cytoprotective function has been described for small heat shock proteins which have disordered N- and C-terminus flanking an alpha-crystalline domain in the middle.” [Page 13. Paragraph 1]*
- *Following the removed part, we added a new citation to our discussion about disordered regions facilitating disaggregation by providing flexible loop regions for disaggregases:*
 - *“It has been proposed that, with amyloid fibers, Hsp70 disaggregase acts through flexible regions to exert pulling forces to the aggregated proteins [84].” [Page 13. Paragraph 1]*

Thank you for sending us your revised manuscript. We have now heard back from the three reviewers who were asked to evaluate your study. As you will see below, while reviewer #3 is overall satisfied with the modifications made, reviewer #1 and #2 still raise a couple of concerns on your work. In light of the reviewers' comment, we would ask you to address the following issues:

1. Reviewer #2's concerns with regards to the immunofluorescence experiments need to be carefully addressed.
2. We understand that addressing reviewer #1's concerns regarding the validation of aggregating proteins could be challenging. The experiments suggested by this reviewer would be welcome, but are not required for acceptance of the manuscript.
3. Reviewer #1' comment on Appendix figure needs to be addressed.

On a more editorial level, please address the following issue.

REFEREE REPORTS

Reviewer #1:

In this revised version the authors updated the manuscript according to some of the points raised in the original review. They performed a lot of changes to the text. Yet, not all suggestions in particular for experiments to gain biological insights were followed though (e.g. depleting chaperones and analysing the disaggregation process; correlation of disaggregation and regain in function of disaggregated proteins, later time points in the disaggregation process).

New comments to the revised manuscript:

Their validation of the aggregating proteins by immunofluorescence does not discriminate between aggregation, protein accumulations and also not between the aggregation of existing proteins or those that are newly synthesised in response to heat shock. Instead they could have performed live cell imaging and e.g. use FRAP or an alternative imaging technique to discriminate between soluble and insoluble protein.

The new figure Appendix 9S is not clear to me. What is depicted on the x-axis? what is the slope in the figure itself?

Reviewer #2:

Thanks to the authors for their work in improving the content of this manuscript. Most of my concerns and the other reviewers' concerns about the experimental design, contextualisation of the findings in light of the literature about heat shock, and mitigating some conclusions were in general addressed.

I have a few additional further comments:

- 1) I find the figure presentation and explanation of the experimental strategy and the definitions of aggregator/soluble protein much clearer now. It is much easier to follow.
- 2) I appreciate the authors' attempt to validate their mass spectrometry data with immunofluorescence (IF) experiments. However, I am not fully convinced that their interpretation of the IF results is fully supported by the data, at least in its current state.
 - A) I would recommend showing the complete array of Hoechst, POI, and merged channels as in 'Appendix figure 6S', instead of showing only the overlay as in 'main figure 3'. This makes it easier to evaluate the cellular localisation of the analyzed proteins.
 - B) For what concerns the IF-BRD4 experiment, I actually disagree with the authors. I do not see an increased intensity of the BRD4 signal in the cytoplasm upon heat shock. The BRD4 signal in the heat shock time course (Appendix figure 6SB) also looks saturated for all points, so it is hard to judge if there was a change over time
 - C) I would say that the TARDBP signal decreases in the nucleus in the heat shock time course (appendix figure 6SB), not that it goes to the cytoplasm.
 - D) I don't find convincing the conclusion that "After five hours of recovery, the cytoplasmic signal of BRD4 and TARDBP was comparable to control level " (lines 301-302) based on the IF images provided in figures 3D and Appendix Figure 6SA.
 - E) In the HELLS experiments the snapshots of the HELLS signal (green channel) at 60 and 120 minutes are swapped (Appendix figure 6SB).Since the authors offer this immunofluorescence experiment to validate their mass spectrometry data, they could improve the presentation of this part of the paper in order to make the conclusions they make out of it more convincing.

Reviewer #3:

The authors have carefully addressed all points I raised. I think this is a nice paper that can now be accepted for publication!

Reviewer #1:

In this revised version the authors updated the manuscript according to some of the points raised in the original review. They performed a lot of changes to the text.

We again thank the reviewer for raising great points during the review that have helped us to improve the manuscript.

Yet, not all suggestions in particular for experiments to gain biological insights were followed though (e.g. depleting chaperones and analysing the disaggregation process; correlation of disaggregation and regain in function of disaggregated proteins, later time points in the disaggregation process).

We appreciate the potential that the reviewer sees for our method. The suggested follow-up experiments would indeed greatly improve our knowledge on protein quality control. In the manuscript, our scope has been on validating the methods we introduce. We hope that the experiments suggested by the reviewer would be the focus of future studies.

New comments to the revised manuscript:

Their validation of the aggregating proteins by immunofluorescence does not discriminate between aggregation, protein accumulations and also not between the aggregation of existing proteins or those that are newly synthesised in response to heat shock. Instead they could have performed live cell imaging and e.g. use FRAP or an alternative imaging technique to discriminate between soluble and insoluble protein.

We agree that techniques, such as FRAP, could discriminate between aggregation and protein accumulation. In addition, there are techniques that can also discriminate between newly synthesized and pre-existing proteins (e.g. photoactivatable fluorescent proteins). We think that foci formation (especially in the case of HELLS) is an indicator of aggregation. Furthermore, for all the proteins we analyzed with immunofluorescence microscopy, the increased protein intensity in the cytoplasm (and foci formation in some cases) correlate with solubility loss in the mass spectrometry experiment - after heat shock and during recovery. As we demonstrate in the manuscript (Figure 4D, Appendix Figure S8A), protein aggregation is very similar in both, newly synthesized and pre-existing protein fraction. Therefore, we did not investigate that aspect in more detail.

Action taken:

- We updated the discussion (additions in bolds and removal in strikethrough):

*“The formation of cytoplasmic foci that coincide with the aggregation detected in mass spectrometry experiment suggests that nuclear proteins do not aggregate in the nucleus, but rather translocate and aggregate in cytoplasm. **It should be noted that the immunofluorescence detection cannot distinguish between aggregate formation and protein accumulation. In addition** However, the link between foci formation and loss of solubility is not trivial.” [Page 13. Paragraph 2]*

The new figure Appendix 9S is not clear to me. What is depicted on the x-axis? what is the slope in the figure itself?

The x-axis shows proteins ordered according to their protein synthesis values (shown on y-axis). The values in the middle of the distribution are very close to each other and the data points form a straight line in the figure (i.e. the slope that we think the reviewer is referring to). We now realize that this is not the best option to illustrate the data and have changed the figure type.

Action taken:

- We have changed the figure presentation to a boxplot that shows also the individual data points and updated the figure legend (see the updated figure below - changes in red)

Appendix figure ~~S9S~~ - Upregulation of Hsp40s (DNAJs) upon heat shock.

~~Proteins are arranged according to protein synthesis level estimated from control-normalized~~
~~intensity of newly synthesized proteins fraction~~ at five hours after recovery from heat shock.
Hsp40s (DNAJs) are highlighted and annotated.

Reviewer #2:

Thanks to the authors for their work in improving the content of this manuscript. Most of my concerns and the other reviewers' concerns about the experimental design, contextualisation of the findings in light of the literature about heat shock, and mitigating some conclusions were in general addressed.

We again thank the reviewer for excellent input during review and appreciate the positive feedback.

I have a few additional further comments:

1) I find the figure presentation and explanation of the experimental strategy and the definitions of aggregator/soluble protein much clearer now. It is much easier to follow.

We are very happy to hear that the reviewer finds the updated manuscript easier to follow.

2) I appreciate the authors' attempt to validate their mass spectrometry data with immunofluorescence (IF) experiments. However, I am not fully convinced that their interpretation of the IF results is fully supported by the data, at least in its current state.

We hope that the revised manuscript now contains convincing interpretation of the immunofluorescence results.

A) I would recommend showing the complete array of Hoechst, POI, and merged channels as in 'Appendix figure 6S', instead of showing only the overlay as in 'main figure 3'. This makes it easier to evaluate the cellular localisation of the analyzed proteins.

We agree with the reviewer and have made the requested changes.

Action taken:

- *The complete image array from Appendix Figure 6SA is now shown in Figure 3D. In addition, the resolution was increased and brightness/contrast adjusted to allow better evaluation of the localization.*
- *The legend of Figure 3D was updated (additions in bold, removals in strikethrough):*

“Protein localization upon heat shock and during recovery as analyzed by immunofluorescence microscopy. HeLa cells were fixed and immunostained with antibodies against indicated target proteins (green) at different conditions [control, after heat shock (10 minutes at 44°C) and after five hours of recovery from the heat shock].

DNA staining (Hoechst) is shown in blue. Magnifications of the boxed areas are shown on top of each image panel. See Appendix Figure 6SA for separate images of target protein and DNA staining. Merged images shown here are the same as in Appendix Figure 6SA. Description of panel A was removed from the legend of Appendix Figure 6SA

- *Title of Appendix Figure 6S was updated (removals in strikethrough):*
“Appendix Figure S6 - Immunofluorescence analysis of protein localization upon heat shock ~~and during recovery.~~”
- *The legend of Appendix Figure S6 was updated (removals in strikethrough):*
“A ~~Cells with or without heat shock (10 minutes at 44°C) as well as after five hours of recovery. Merged images shown here are the same as in Figure 3D.~~
B ~~Cells after exposure to heat shock (at 44°C) for 0, 10, 20, 60 or 120 minutes.”~~
- *References in the text were updated (i.e. Appendix Figure S6A is now Figure 3D and Appendix figure S6B is now Appendix Figure S6)*

B) For what concerns the IF-BRD4 experiment, I actually disagree with the authors. I do not see an increased intensity of the BRD4 signal in the cytoplasm upon heat shock. The BRD4 signal in the heat shock time course (Appendix figure 6SB) also looks saturated for all points, so it is hard to judge if there was a change over time

We agree that the BRD4 signal in the cytosol after heat shock is not very strong - especially when comparing to HELLS. However, we observed a clear increase in the cytoplasmic signal upon heat shock. This is now also more clear because we provide the figure at higher resolution, in the previous submission and entirely through our own fault we used a figure where the resolution was lower, we apologies for that. The increased signal can be seen in two independent experiments (heat shock with recovery and with the time-series analysis of heat shock). We hope that this is now more apparent in the updated Figure 3D.

We agree that the signal is saturated for BRD4. However, we think it is unlikely that signal saturation in cytoplasm could explain why we repeatedly detect increase in cytoplasm upon heat shock. In addition, the signal decreases during recovery from heat shock. From these observations, we are convinced that the increased cytoplasmic BRD4 signal upon heat shock is not because of signal saturation.

Action taken:

- *Appendix Figure S6A with higher resolution and adjusted brightness/contrast is moved to Figure 3D.*

C) I would say that the TARDBP signal decreases in the nucleus in the heat shock time course (appendix figure 6SB), not that it goes to the cytoplasm.

In the manuscript, we stated that TARDBP forms nuclear foci but do not mention about translocation to cytoplasm since we also did not make this observation. We agree that the images could be interpreted as decrease in nuclear signal. The foci could be present without the heat shock and would become visible after non-foci signal decreased.

Action taken:

- We updated the text (additions in bold):

*“The cytoplasmic BRD4 formed bigger foci during prolonged heat shock while TARDBP formed nuclear foci **or the intensity from non-foci proteins decreased** (Appendix Figure S6).” [Page 7. Paragraph 5]*

D) I don't find convincing the conclusion that "After five hours of recovery, the cytoplasmic signal of BRD4 and TARDBP was comparable to control level " (lines 301-302) based on the IF images provided in figures 3D and Appendix Figure 6SA.

Based on the comment 1B we understand that evaluating the cytoplasmic signal was not so straightforward. However, the decrease in cytoplasmic signal during recovery should be evident, especially from the updated Figure 3D. We removed the comparison to control condition in our conclusion and now just state that the cytoplasmic signal was decreased.

Action taken:

- We have updated the text (additions in bold and removals in strikethrough):

*“After five hours of recovery, **a reduction in** the cytoplasmic signal of BRD4 and TARDBP was **observed** ~~comparable to control level~~, while the cytoplasmic HELLS remained in foci” [Page 7. Paragraph 6]*

E) In the HELLS experiments the snapshots of the HELLS signal (green channel) at 60 and 120 minutes are swapped (Appendix figure 6SB).

We are very thankful for the reviewer for spotting the mistake and we have now fixed it.

Action taken:

- Images of HELLS signal at 60 and 120 minutes after heat shock are swapped back

Since the authors offer this immunofluorescence experiment to validate their mass spectrometry data, they could improve the presentation of this part of the paper in order to make the conclusions they make out of it more convincing.

We hope that the changes discussed above have improved the presentation.

Reviewer #3:

The authors have carefully addressed all points I raised. I think this is a nice paper that can now be accepted for publication!

We again thank the reviewer for all the excellent comments that we think have improved the manuscript greatly and appreciate the positive feedback.

Corresponding Author Name: Mikhail M Savitski

Manuscript Number: MSB-20-9500